

# Wegner's Ising gauge spins versus Kitaev's Majorana partons: Mapping and application to anisotropic confinement in spin-orbital liquids

Urban F. P. Seifert[1] and Sergej Moroz[2,3]⋆

**1** Kavli Institute of Theoretical Physics, University of California,
Santa Barbara, California 93106, USA
**2** Department of Engineering and Physics, Karlstad University, Karlstad, Sweden
**3** Nordita, KTH Royal Institute of Technology and Stockholm University, Stockholm, Sweden

⋆ urbanseifert@ucsb.edu

## Abstract

Emergent gauge theories take a prominent role in the description of quantum matter, supporting deconfined phases with topological order and fractionalized excitations. A common construction of $\mathbb{Z}_2$ lattice gauge theories, first introduced by Wegner, involves Ising gauge spins placed on links and subject to a discrete $\mathbb{Z}_2$ Gauss law constraint. As shown by Kitaev, $\mathbb{Z}_2$ lattice gauge theories also emerge in the exact solution of certain spin systems with bond-dependent interactions. In this context, the $\mathbb{Z}_2$ gauge field is constructed from Majorana fermions, with gauge constraints given by the parity of Majorana fermions on each site. In this work, we provide an explicit Jordan-Wigner transformation that maps between these two formulations on the square lattice, where the Kitaev-type gauge theory emerges as the exact solution of a spin-orbital (Kugel-Khomskii) Hamiltonian. We then apply our mapping to study local perturbations to the spin-orbital Hamiltonian, which correspond to anisotropic interactions between electric-field variables in the $\mathbb{Z}_2$ gauge theory. These are shown to induce anisotropic confinement that is characterized by emergence of weakly-coupled one-dimensional spin chains. We study the nature of these phases and corresponding confinement transitions in both absence and presence of itinerant fermionic matter degrees of freedom. Finally, we discuss how our mapping can be applied to the Kitaev spin-1/2 model on the honeycomb lattice.



# 1 Introduction

Discrete gauge fields play a prominent role in various branches of physics. Discovered more than fifty years ago by Wegner [1], the Ising gauge theory was the first and simplest example of a lattice gauge theory that predated the general construction of lattice gauge theories by Wilson [2]. Remarkably, the two phases of this model cannot be distinguished by a local order parameter, but instead they can be diagnosed by the decay behavior of extended Wegner-Wilson gauge-invariant loop operators. More generally, the model in its deconfined phase exhibits $\mathbb{Z}_2$ topological order which is robust under arbitrary small perturbations [3,4]. In the absence of a tension in electric strings, the pure $\mathbb{Z}_2$ gauge theory is equivalent to the toric code with a fixed charge configuration [5], an exactly solvable model that formed a paradigm of topological quantum computing. Currently, the Ising gauge theory is our best means for understanding the intricate phenomenology of gapped $\mathbb{Z}_2$ spin liquids [6,7] emerging, for example, in cold atom platforms [8–10]. Coupling of two-dimensional quantum Ising gauge fields to dynamical matter revealed new phases of matter [11–18] and exotic bulk and boundary phase transitions [19–26]. The quest for engineering the Wegner Ising gauge theory with quantum technologies is ongoing [27–39].

A different incarnation of the Ising gauge theory was discovered by Kitaev [40] in his by now famous exactly-solvable honeycomb model. Here the link Ising gauge fields are assembled from Majorana partons that fractionalize microscopic spin-1/2 degrees of freedom residing on sites. In the Kitaev model, Ising gauge fields are static and couple to one remaining itinerant Majorana fermion that carries a $\mathbb{Z}_2$ charge. Recently, Kitaev's approach was generalized to models with more degrees of freedom per site [41–46]. These models naturally involve static $\mathbb{Z}_2$ gauge fields coupled to multiplets of $\nu$ itinerant Majoranas that enjoy an SO($\nu$) global internal symmetry.

One may wonder if and how the Wegner's and Kitaev's incarnations of the Ising gauge theory are related to each other. To answer this question, in this paper, we investigate the Ising gauge theory coupled to dynamical single-component complex fermion matter which represents the $\nu = 2$ spin-orbital liquid. First, we demonstrate how the conventional bosonic formulation of the model, where the Ising gauge operators are represented by spin-1/2 Pauli matrices acting on links, is related to the Kitaev-inspired fermionic formulation, where the gauge fields are built of Majorana partons. More specifically, we develop an explicit mapping between the two formulations based on a carefully constructed non-local Jordan-Wigner transformation. This allows us to demonstrate that the local Gauss laws in those two formulations are identical to each other. Moreover, we discuss subtleties arising from the presence of boundaries, where the Jordan-Wigner string ends. Our non-local transformation is an addition to a plethora of higher-dimensional bosonization duality mappings investigated recently by various authors [47–59], for related earlier works see [60–65].

Can our mapping be used for the original Kitaev model? While our mapping is not directly applicable on the honeycomb lattice, the honeycomb Kitaev model can be equivalently rewritten as a $\mathbb{Z}_2$-gauged p-wave superconductor on a square lattice, see Appendix A. Our Jordan-Wigner mapping can now be straightforwardly applied to this formulation of Kitaev's spin-1/2 honeycomb model.

Apart from conceptual appeal, this explicit non-local mapping allows one to express local non-integrable[1] perturbations of generalized Kitaev spin-orbital models as non-standard flux-changing electric terms in the Ising lattice gauge theory. Conversely, the mapping may also be used in reverse to analyse new non-trivial perturbations of spin-orbital liquids. In this paper, the above-mentioned mapping is employed to investigate and elucidate anisotropic confinement, wherein fractionalized excitations are dimensionally imprisoned: they are free to propagate without string tension along certain one-dimensional chains while becoming confined in the direction transverse to those chains.[2]

The remainder of this work is structured as follows. In Sec. 2, we introduce Wegner's formulation of $\mathbb{Z}_2$ lattice gauge theory with fermionic matter and review the exact solution of a spin-orbital model on the square lattice based on Kitaev's Majorana parton construction. In Sec. 3, we construct the mapping between Wegner's and Kitaev's constructions, and discuss subtleties arising from the presence of boundaries. Sec. 4 is concerned with the application of our mapping to anisotropic confinement in spin-orbital liquids. The paper is concluded in Sec. 5.

---

[1]We define non-integrable perturbations as the ones that introduce dynamics to the $\mathbb{Z}_2$-gauge field.

[2]This is somewhat reminiscent to the effectively one-dimensional dispersion of anyons that emerges in the (gapped) anisotropic Kitaev model perturbed by a Zeeman field as discussed in Ref. [66].

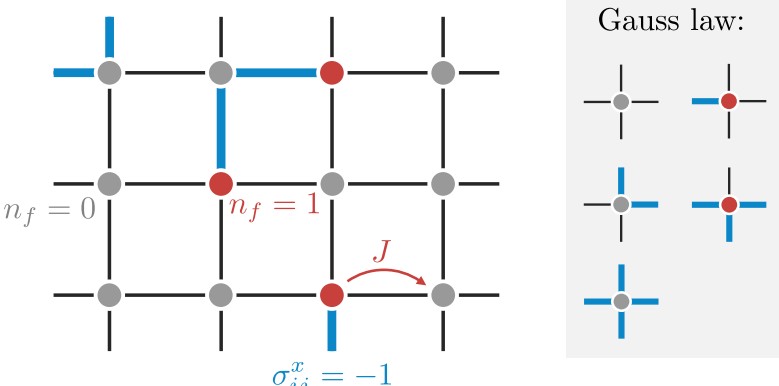

Figure 1: Illustration of $\mathbb{Z}_2$ gauge theory with spinless (complex) fermionic matter on the square lattice: Wegner gauge spins are located on bonds, with electric strings ($\sigma_{ij}^x = -1$) indicated by blue thick lines. Fermions (indicated in red) occupy sites of the lattice. Configurations allowed by the $\mathbb{Z}_2$ Gauss law are shown in the right panel. Fermions can hop with the coupling $J$ but must remain connected by electric field strings so that the Gauss law is satisfied.

## 2  $\mathbb{Z}_2$ lattice gauge theories with complex fermionic matter

### 2.1  Conventional lattice gauge theory bosonic formulation

We start from a two-dimensional $\mathbb{Z}_2$ lattice gauge theory defined on links of a square lattice that is coupled to itinerant complex single-component fermions that reside on sites of the lattice, see Fig. 1.

Quantum dynamics of the model is governed by the Hamiltonian

$$\mathcal{H}_{\text{LGT}} = -J \sum_{\langle ij \rangle} \left( f_i^\dagger \sigma_{ij}^z f_j + \text{h.c.} \right) - J_\square \sum_\square \prod_{\langle ij \rangle \in \square} \sigma_{ij}^z \,. \tag{1}$$

Here $i$ and $j$ label sites of the lattice and $\langle ij \rangle$ denotes a nearest-neighbour pair of sites. The coupling $J$ controls the hopping amplitude of the gauge-charged fermion $f$, while $J_\square$ assigns energy to the elementary magnetic plaquettes

$$W_\square = \prod_{l \in \square} \sigma_l^z \,. \tag{2}$$

Clearly, the Ising gauge fields are static because the Hamiltonian contains no terms that do not commute with $\sigma^z$ at any link. As a result, the Hamiltonian commutes with each elementary plaquette operator such that the ground-state sector is given by those $W_\square = \pm 1$ quantum numbers which minimize the ground-state energy. This extensive number of conserved quantities makes the problem integrable and thus allows for the exact solution for both the ground state and excitations.

The Hamiltonian is invariant under *discrete gauge transformations* $f_i \rightarrow s_i f_i$, and $\sigma_{ij}^z \rightarrow s_i \sigma_{ij}^z s_j$ where $s_i = \pm 1 \in \mathbb{Z}_2$, which are generated by the operator

$$G_j = (-1)^{f_j^\dagger f_j} \prod_{l \in +_j} \sigma_l^x \,. \tag{3}$$

Noting that $[\mathcal{H}_{\text{LGT}}, G_j] = 0 \forall j$, we see that the complete Hilbert space then decomposes into distinct superselection sectors with eigenvalues $G_i = \pm 1$. In the absence of any fermions,

$G_i = -1$ implies that an odd number of electric strings terminates at site $i$, which is analogous to an electric field string ending at an electromagnetic charge. $G_i = -1$ hence corresponds to a $\mathbb{Z}_2$ background charge placed at site $i$. In the following, we want to consider the case that all charges in the theory are *dynamical* and carried by the fermions $f$, so that we will work with a homogeneous background which satisfies the constraint $G_j = +1 \forall j$.

In addition, the model enjoys several global symmetries: First, we have a $U(1)$ particle number symmetry that acts only on fermions as $f \to e^{i\alpha} f$. Second, the Hamiltonian and the Gauss constraint are invariant under the $D_4$ point group and discrete single-link translations. Third, an anti-unitary time-reversal symmetry acts as complex conjugation on Ising gauge field Pauli matrices (i.e. $\sigma^y \to -\sigma^y$), but leave fermions invariant. Finally, one can construct a particle-hole symmetry: to this this end, one starts from the transformation $f_i \to (-1)^i f_i^\dagger$ with the checkerboard pattern $(-1)^i = (-1)^{i_x + i_y}$. Given that only the nearest-neighbour hopping is present in the Hamiltonian Eq. (1), the operator implementing this transformation commutes with $\mathcal{H}_{\text{LGT}}$. Note, however, that on its own this transformation violates the Gauss constraint since under this transformation $n_i = f_i^\dagger f_i \to 1 - n_i$ and thus $G_j \to -G_j$. We can fix this problem by additionally flipping an odd number of electric strings operators attached to each site. One simple implementation is to flip one (for example the north) string adjacent to every site of the A-sublattice with $\sigma^x \to \sigma^z \sigma^x \sigma^z = -\sigma^x$. The combined transformation preserves the Gauss law and commutes with the Hamiltonian, so it is a symmetry. Curiously, a product of different implementations of this particle-hole symmetry gives rise to a set of closed Wegner-Wilson loop operators $W_\mathcal{C} = \prod_{l \in \mathcal{C}} \sigma_l^z$ which, obviously, commute with the Hamiltonian in Eq. (1) and the Gauss law [Eq. (3)]. The full set of these loop operators generates a one-form magnetic symmetry [67, 68].

While the model in Eq. (1) is expressed in terms of redundant gauge-variant degrees of freedom, it has been demonstrated in Refs. [24, 31, 32] using a local mapping that it is completely equivalent to a model of spin 1/2 gauge-invariant degrees of freedom acting on links of a square lattice.

## 2.2 Spin-orbital liquids and their fermionic $\mathbb{Z}_2$ gauge-redundant formulation

We now show how a different formulation of the Ising lattice gauge theory coupled to complex fermionic matter emerges naturally in the exact solution of generalized Kitaev models. To this end, we first look at the $\nu = 2$ generalization of the Kitaev model on the square lattice, which can be written in terms of two SU(2) degrees of freedom on each site, with Pauli matrices $\mathsf{s}^\alpha, \tau^\beta$, respectively [44–46]. These degrees of freedom can, for example, correspond to spin and and orbital degrees of freedom in Kugel-Khomskii-type models. Denoting $\tau^\mu = (\tau^x, \tau^y, \tau^z, \mathbb{1})$, the Hamiltonian contains a biquadratic interaction involving bond-dependent interactions in the orbital sector and U(1)-symmetric XY exchange in the spin sector,

$$\mathcal{H}_{\text{SOL}} = -K \sum_{\langle ij \rangle_\mu} \left( \mathsf{s}_i^x \mathsf{s}_j^x + \mathsf{s}_i^y \mathsf{s}_j^y \right) \tau_i^\mu \tau_j^\mu. \tag{4}$$

Generalizing Kitaev's solution of the honeycomb model [40], the model can be solved exactly be representing the spin-orbital operators (which furnish a representation of the Clifford algebra and can be rewritten in terms of $4 \times 4$-dimensional Gamma matrices [45]) via 6 Majorana fermions $b^\mu, c^x, c^y$ with $\{b^\mu, b^\nu\} = 2\delta_{\mu\nu}$ and similarly for $c^x, c^y$. In particular, we identify

$$\mathsf{s}^\alpha = \frac{-\mathrm{i}}{2} \begin{pmatrix} c^x \\ c^y \\ b^4 \end{pmatrix} \times \begin{pmatrix} c^x \\ c^y \\ b^4 \end{pmatrix}, \quad \text{and} \quad \tau^\alpha = \frac{-\mathrm{i}}{2} \epsilon^{\alpha\beta\gamma} b^\beta b^\gamma, \tag{5}$$

which satisfy the SU(2) commutation relations and keep the identities $\mathsf{s}^x \mathsf{s}^y \mathsf{s}^z = \mathrm{i} = \tau^x \tau^y \tau^z$ manifest. Since 6 Majorana fermions form a 8-dimensional Hilbert space, but the spin-orbital

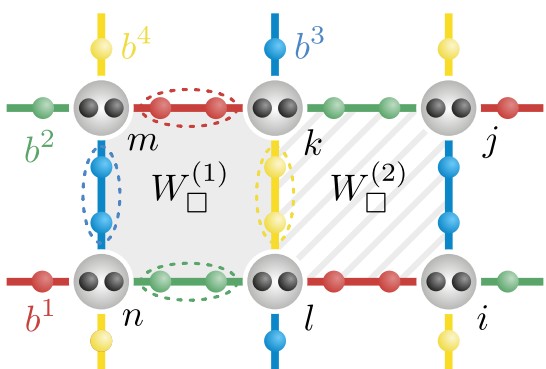

Figure 2: Illustration of the emergent $\mathbb{Z}_2$ gauge fields: Type-(1,2,3,4) links are colored (red, green, blue, yellow), and the $b^1, \ldots, b^4$ Majorana fermions on each site can be used to define the $\mathbb{Z}_2$ gauge field $u_{ij} = i b_i^\mu b_j^\mu$ on type-$\mu$ links. Solid (hatched) grey shaded plaquettes indicate the plaquette operators $W_\square^{(1)}$ and $W_\square^{(2)}$, respectively.

SU(2)×SU(2) representation is only four-dimensional, the Majorana representation introduces redundant states. The redundancy is removed by enforcing a local fermion parity constraint $D_j \equiv q_j \in \mathbb{Z}_2$, where $D_j$ is the local fermion parity operator

$$D_j = (-i) b_j^1 b_j^2 b_j^3 b_j^4 c_j^x c_j^y , \tag{6}$$

with eigenvalues $\pm 1$, which are fourfold degenerate each (see also [69,70]). Choosing $q_j = \pm 1$ and then enforcing the constraint thus projects out four redundant states. Using the parity operator in Eq. (6), and enforcing the fermion parity constraint, the combined spin-orbital operators can be written as

$$\mathsf{s}^\beta \tau^\alpha = i D b^\alpha c^\beta \equiv i q b^\alpha c^\beta , \tag{7}$$

where $\alpha = 1, 2, 3$, $\beta = x, y$ and $q \in \mathbb{Z}_2$. Similarly, one finds that

$$\mathsf{s}^z \tau^x = i D b^1 b^4 , \quad \mathsf{s}^z \tau^y = i D b^2 b^4 , \quad \text{and} \quad \mathsf{s}^z \tau^z = i D b^3 b^4 , \tag{8}$$

where again in any particular chosen sector, the local fermion parity $D$ can be replaced by its eigenvalue $q = \pm 1 \in \mathbb{Z}_2$. Using the representation in Eq. (5), the Hamiltonian then becomes

$$\mathcal{H}_{\text{SOL}} = K \sum_{\langle ij \rangle_\alpha} i q_i q_j b_i^\alpha b_j^\alpha \left( i c_i^x c_j^x + i c_i^y c_j^y \right) + K \sum_{\langle ij \rangle_4} i b_i^4 b_j^4 \left( i c_i^x c_j^x + i c_i^y c_j^y \right) . \tag{9}$$

Analogous to Kitaev's exact solution, the pairs of $b$-type Majorana fermions form a $\mathbb{Z}_2$ gauge field, given by $u_{ij} = i q_i q_j b_i^\alpha b_j^\alpha$ on $\alpha = 1, 2, 3$-links, and $u_{ij} = i b_i^4 b_j^4$ on 4-links, see Fig. 2 for an illustration. Without loss of generality, we choose $q = +1$ in the following. Here, $\mathbb{Z}_2$ gauge transformations are generated by the fermion parity operators, since $D_i u_{ij} = -u_{ij} D_i$. On the other hand, the $c^x, c^y$ Majorana fermions are charged under the $\mathbb{Z}_2$ gauge field and can be combined into one complex spinless fermion $f_i = (c_i^x + i c_i^y)/2$. Then, the fermion parity constraint reads

$$D_j = b_j^1 b_j^2 b_j^3 b_j^4 (-1)^{f_j^\dagger f_j} \equiv q_j . \tag{10}$$

In the following, for notational convenience, we redefine fermions on the B sublattice $f_j \mapsto i f_j$ such that the Hamiltonian is written as

$$\mathcal{H}_{\text{SOL}} = -2K \sum_{\langle ij \rangle} (f_i^\dagger u_{ij} f_j + \text{h.c.}) . \tag{11}$$

From $\{D_i, u_{ij}\} = 0$ it is seen that the constraint operator $D_j$ generates $\mathbb{Z}_2$ gauge transformations of $u_{ij} \rightarrow s_i u_{ij} s_j$ and $f_j \rightarrow s_j f_j$ with $s_j = \pm 1 \in \mathbb{Z}_2$. As in Kitaev's honeycomb model [40], the exact solution of $\mathcal{H}_{\text{SOL}}$ is enabled by the presence of an extensive number of gauge-invariant conserved quantities $W_\square = \prod_{\langle ij \rangle \in \square} u_{ij}$ with eigenvalues $\pm 1$.

Note that while the spin-orbital Hamiltonian in Eq. (4) naively appears to break the elementary lattice symmetries of the square lattice. Notwithstanding, we can define generalized spatial symmetries which combine the elementary spatial transformations of a square lattice with spin-orbital operations. For example, the system is invariant under translations $\boldsymbol{m}_1 = \hat{x}$ and $\boldsymbol{m}_2 = \hat{y}$ if paired together with spin-orbital rotations that interchange interactions on bond types $1 \leftrightarrow 2$ and $3 \leftrightarrow 4$. These spin-orbital operations can be rewritten in terms of the SO(4) group acting on the vector of four-dimensional Gamma matrices $\Gamma^\mu$ where $\mu = 1, \ldots, 4$ [45]. The particle-hole symmetry has also been identified in [45].

Clearly, the Hamiltonian Eq. (11) can be seen to be equivalent to the Hamiltonian in Eq. (1) at $J_\square = 0$ and $J \equiv 2K$ upon identifying the gauge fields $\sigma^z_{ij}$ and $u_{ij}$. Both Hamiltonians describe fermions hopping in the background of a $\mathbb{Z}_2$ gauge field on the square lattice. However, it is not immediately clear how the generators of the Ising gauge transformations in the two formulations, i.e. the fermion parity constraint operator $D_j$ in Eq. (6) and the vertex Gauss law operator $G_j$ in Eq. (3), are related. Constructing an explicit mapping between these two constraints will be a main goal of Sec. 3.

Finally we notice that the gauge-invariant plaquette operators $W_\square$, which individually commute with $\mathcal{H}_{\text{SOL}}$, can be straightforwardly expressed in the spin-orbital language. As becomes clear from Fig. 2, there are two types of plaquettes in the spin-orbital formulation. Focusing on the plaquette of type (1), we can write

$$
\begin{aligned}
W_\square^{(1)} = \prod_{l \in \square} u_l &= (ib^1_m b^1_k)(ib^4_l b^4_k)(ib^2_l b^2_n)(ib^3_m b^3_n) \\
&= (ib^1_m b^3_m)(ib^1_k b^4_k)(ib^4_l b^2_l)(ib^2_n b^3_n) = (s^z_k s^z_l)(\tau^y_m \tau^x_k \tau^y_l \tau^x_n),
\end{aligned}
\tag{12}
$$

where we have used Eqs. (5) and (8) to rewrite the site-local Majorana-fermion bilinears in terms of spin-orbital operators. One may proceed similarly for plaquettes of second type, obtaining the representation

$$
W_\square^{(2)} = (s^z_k s^z_l)(\tau^x_l \tau^y_k \tau^x_j \tau^y_i).
\tag{13}
$$

While the two types of plaquettes appear differently, the generalized translation transformation maps them to each other.

## 3 Mapping

As noted in the previous section, the $\mathbb{Z}_2$ lattice gauge theory that emerges upon rewriting Kitaev-type spin-orbital models using the Majorana parton description [see Eq. (11)] can be directly related to Wegner's formulation of the $\mathbb{Z}_2$ lattice gauge theory upon identifying the Majorana gauge field $u_{ij} = ib^\mu_i b^\mu_j$ on $\mu = \langle ij \rangle$ links with the Ising gauge field $\sigma^z_{ij}$ in Eq. (1).

A crucial remaining task is to find an explicit relation between the Majorana fermion parity constraint operator $D_j$ in Eq. (6), which involves only degrees of freedom on site $j$, and the $\mathbb{Z}_2$ Gauss law in Eq. (3), which additionally involves degrees of freedom placed on *bonds* emanating site $j$. In order to establish an explicit correspondence we thus must answer the following question: *How is the electric field $\sigma^x_{ij}$ related to the gauge Majorana fermions $b^\mu_j$ in Kitaev-type $\mathbb{Z}_2$ lattice gauge theories?*

We will show in this section that a carefully constructed Jordan-Wigner transformation provides an explicit mapping between Kitaev-type and more conventional Wegner-type gauge theories.

To that end, we first note that on a given $\langle ij \rangle_\mu$-link of type $\mu = 0, \ldots, 3$, the associated gauge Majorana fermions $b_i^\mu$, $b_j^\mu$ can be combined into a single (spinless) complex *bond* fermion $d_{ij} = (b_i^\mu + i b_j^\mu)/2$, where we pick the convention $i \in$ A and $j \in$ B sublattices. As discussed earlier, we identify the gauge field $u_{ij}$ in Eq. (11) with $\sigma_{ij}^z$ in Eq. (1), such that the spin-orbital liquid Hamiltonian Eq. (11) maps onto the fermionic part of the Hamiltonian for $\mathbb{Z}_2$ LGT in Wegner's formulation as in Eq. (1),

$$\mathcal{H}_{\text{SOL}} = -2K \sum_{\langle ij \rangle} u_{ij} \left( f_i^\dagger f_j + \text{h.c.} \right) \mapsto \tilde{\mathcal{H}}_{\text{SOL}} = -2K \sum_{\langle ij \rangle} \sigma_{ij}^z \left( f_i^\dagger f_j + \text{h.c.} \right). \tag{14}$$

This identification implies that we can write

$$\sigma_{ij}^z \equiv u_{ij} = i b_i^\mu b_j^\mu = 2 d_{ij}^\dagger d_{ij} - 1, \tag{15}$$

which, crucially, can be understood as the fermionic representation of the bond-local Pauli matrix $\sigma_{ij}^z$. This is the key insight underlying our mapping. In the following, we will argue that the corresponding transverse operators $\sigma_{ij}^x = \sigma_{ij}^+ + \sigma_{ij}^-$ and $\sigma_{ij}^y = (\sigma_{ij}^+ - \sigma_{ij}^-)/i$ are obtained using a Jordan-Wigner relation,

$$\sigma_{ij}^+ = \exp\left[ -i\pi \sum_{l < \langle ij \rangle} d_l^\dagger d_l \right] d_{ij}^\dagger, \quad \text{and} \quad \sigma_{ij}^- = \exp\left[ i\pi \sum_{l < \langle ij \rangle} d_l^\dagger d_l \right] d_{ij}, \tag{16}$$

where the sum extends over all links $l$ up to (but not including) the link $\langle ij \rangle$ along a one-dimensional path through the two-dimensional lattice which enumerates (and defines a unique ordering of) all bonds. By construction, $\sigma_l^z$ and $\sigma_{l'}^\pm$ satisfy the SU(2) commutation relations *iff* $l = l'$, and commute otherwise. While in principle any choice of a path is conceivable, they typically lead to Jordan-Wigner transformations under which local interactions become non-local, such that the Jordan-Wigner transformation for systems with spatial dimension $d > 1$ has often been considered impractical. However, for the system at hand, we show that a particular snake-like diagonal path $\mathcal{P}$, as shown in Fig. 3 leads to an invertible transformation between $\mathcal{H}_K$ and $\mathcal{H}_{\text{LGT}}$, and moreover relates the fermion parity operator $D_j$ to the $\mathbb{Z}_2$ Gauss law $G_j$.

We first discuss the mapping in the bulk, and then later on consider the mapping in the presence of a boundary.

## 3.1 Bulk mapping

We first note that we can write Jordan-Wigner phase operator as

$$\exp\left[ \mp i\pi \sum_{l < \langle ij \rangle} d_l^\dagger d_l \right] = \prod_{l < \langle ij \rangle} (-1)^{d_l^\dagger d_l} = \prod_{l < \langle ij \rangle} (-u_l), \tag{17}$$

where we have used $(-1)^{d^\dagger d} = 1 - 2 d^\dagger d$, thus with Eq. (16) we can write

$$\sigma_{ij}^x = \prod_{l < \langle ij \rangle_\mu} (-u_l) b_i^\mu, \quad \text{and} \quad \sigma_{ij}^y = - \prod_{l < \langle ij \rangle_\mu} (-u_l) b_j^\mu, \tag{18}$$

where $\langle ij \rangle_\mu$ is a link of type $\mu$, and $i \in$ A, $j \in$ B sublattice. The inverse mapping is readily constructed as

$$b_i^\mu = \prod_{l < \langle ij \rangle_\mu} \left( -\sigma_l^z \right) \sigma_{ij}^x, \quad \text{and} \quad b_j^\mu = - \prod_{l < \langle ij \rangle_\mu} \left( -\sigma_l^z \right) \sigma_{ij}^y, \tag{19}$$

where we take $i \in$ A and $j \in$ B sublattices.

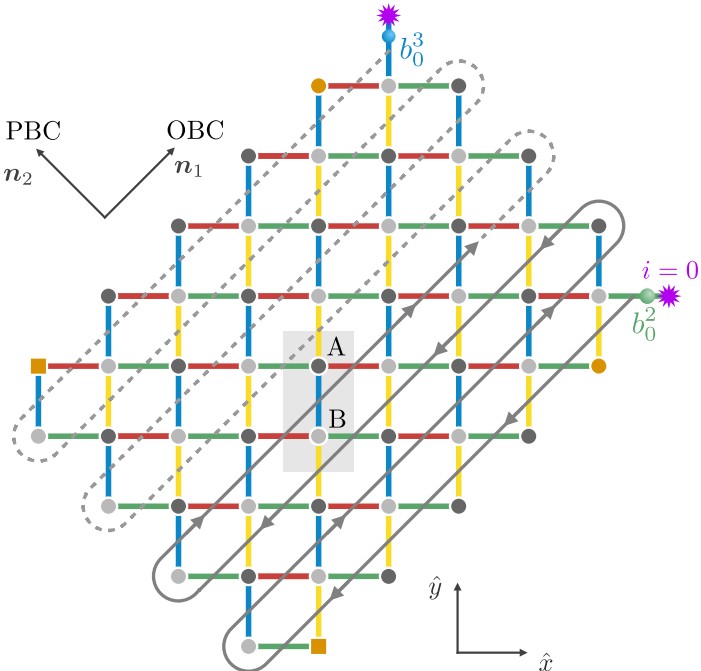

Figure 3: Illustration of the Jordan-Wigner string (shown in dark grey) for defining an ordering of bonds on a cylindrical geometry, with periodic boundary conditions (PBC) along $n_2$ (the orange squares/circles indicates two examples of sites which are identified) and open boundary conditions (OBC) along $n_1$. We mark the first site $i = 0$ by a purple star.

### 3.1.1 Gauge-invariant electric field operators

We seek an expression for the *gauge-invariant* electric field operators in lattice $\mathbb{Z}_2$ gauge theory in terms of the above Jordan-Wigner construction. Note that the right-hand side of Eq. (18) is composed of the gauge field $u_{ij}$ and gauge Majoranas $b_i^\mu$, which transform non-trivially under gauge transformations. To identify the electric field operators, we first note the behaviour of Jordan-Wigner string operator up to the link $\langle ij \rangle_\mu$ under gauge transformations generated by $D_i$ *in the bulk* (neglecting for now boundary conditions, and the first bond in the Jordan-Wigner string).

On type-1 (type-2) bonds, $\langle i, i \pm \hat{x} \rangle$, with $i \in$ A sublattice, Fig. 3 makes obvious that under general gauge transformations acting on all sites $u_{ij} \mapsto s_i u_{ij} s_j$ with $s_i = \pm 1 \in \mathbb{Z}_2$, one has

$$\left[ \prod_{l < \langle i, i \pm \hat{x} \rangle_{1(2)}} (-u_l) \right] = \left[ \ldots (-u_{i \mp \hat{x} \mp \hat{y}, i \mp \hat{y}})(-u_{i, i \mp \hat{y}}) \right] \rightarrow s_i \left[ \prod_{l < \langle i, i \pm \hat{x} \rangle_{1(2)}} (-u_l) \right], \quad (20)$$

because each site index $j$ (except for i) occurs twice in expanding the product, and thus any element of the $s_j \in \mathbb{Z}_2$ gauge group $(s_j)^2 = (\pm 1)^2 = 1$ cancels, except at site $i$. For JW-strings terminating on type-3 (type-4) bonds $\langle i, i \mp \hat{y} \rangle$, we have

$$\left[ \prod_{l < \langle i, i \mp \hat{y} \rangle_{3(4)}} (-u_l) \right] = \left[ \ldots (-u_{i \mp \hat{x} 2 \mp \hat{y}, i \mp \hat{x} \mp \hat{y}})(-u_{i \mp \hat{x} \mp \hat{y}, i \mp \hat{y}}) \right] \rightarrow s_{i \mp \hat{y}} \left[ \prod_{l < \langle i, i \mp \hat{y} \rangle_{3(4)}} (-u_l) \right]. \quad (21)$$

Single gauge Majoranas transform under gauge transformations as $b_j^\mu \rightarrow s_j b_j^\mu$ for any site $j$ and $s_j = \pm 1$. Given Eqs. (20) and (21), we can therefore conclude that on type-1 (type-2) links

$\langle i, i \pm \hat{x} \rangle$, the expression for operator $\sigma^x_{i,i\pm\hat{x}}$ as defined in Eq. (18) is invariant under (bulk) gauge transformations generated by the local fermion parity $D_j$, while on type-3 (type-4) links $\langle i, i \mp \hat{y} \rangle$ the operator $\sigma^y_{i,i\mp\hat{y}}$ is invariant under $\mathbb{Z}_2$ transformations generated by the fermion parity constraint.

Since $\sigma^x_{i,i\pm\hat{x}}$ and $\sigma^y_{i,i\pm\hat{y}}$ are gauge invariant, they can again be expressed in terms of gauge-invariant spin-orbital operators. To this end, one may expand the product over the string operator in Eq. (18), insert $u_{ij} = ib^\mu_i b^\mu_j$ and then reorder Majorana fermions (in every second factor) such that equal-site gauge Majoranas are grouped into pairs, for example (note that $i \in A$ sublattice)

$$
\begin{aligned}
\sigma^x_{i,i+\hat{x}} &= \dots (-u_{i-2\hat{x}-2\hat{y},i-\hat{x}-2\hat{y}})(-u_{i-\hat{x}-\hat{y},i-\hat{x}-2\hat{y}})(-u_{i-\hat{x}-\hat{y},i-\hat{y}})(-u_{i,i-\hat{y}})b^1_i \\
&= \dots (-ib^1_{i-2\hat{x}-2\hat{y}}b^1_{i-\hat{x}-2\hat{y}})(ib^3_{i-\hat{x}-2\hat{y}}b^3_{i-\hat{x}-\hat{y}})(ib^3_{i-\hat{x}-2\hat{y}}b^3_{i-\hat{x}-\hat{y}})(-ib^1_{i-\hat{x}-\hat{y}}b^1_{i-\hat{y}})(ib^3_{i-\hat{y}}b^3_i)b^1_i \\
&= \dots (-\tau^y_{i-\hat{x}-2\hat{y}})(-\tau^y_{i-\hat{x}-\hat{y}})(-\tau^y_{i-\hat{y}})(-\tau^y_i),
\end{aligned}
\tag{22}
$$

where the last equation follows from rewriting equal-site Majorana pairs in terms of the spin-orbital operators using Eq. (5). Note that above, we have only explicitly written down terms due to red-blue bonds on the string operator (ending on site $i$). A similar procedure follows for green-yellow links, where one may write, for example,

$$
\begin{aligned}
\sigma^y_{i,i+\hat{y}} &= -\Big[ \dots (-u_{i+2\hat{x}+2\hat{y},i+\hat{x}+2\hat{y}})(-u_{i+\hat{x}+\hat{y},i+\hat{x}+2\hat{y}})(-u_{i+\hat{x}+\hat{y},i+\hat{y}})b^4_{i+\hat{y}} \Big] \\
&= -(\dots)(-s^z_{i+\hat{x}+2\hat{y}}\tau^y_{i+\hat{x}+2\hat{y}})(-s^z_{i+\hat{x}+\hat{y}}\tau^y_{i+\hat{x}+\hat{y}})(-s^z_{i+\hat{y}}\tau^y_{i+\hat{y}}).
\end{aligned}
\tag{23}
$$

Note that, as visible from Fig. 3, the string operator will in general contain both blue-red and green-yellow segments. If all bonds emanating from a given site $j$ are traversed by the string operator, above calculation shows that both factors $(-\tau^y_j)$ and $(-s^z_j\tau^y_j)$ will be contained in the string operator, which can be simplified to $(-\tau^y_j)(-s^z_j\tau^y_j) \equiv s^z_j$. However, this does not hold for boundary sites (which lead to factors of $\tau^x$ in the expanded and reordered string operators) or sites which only have two emanating bonds traversed by the string operator. Therefore, no *simple* general closed-form expressions of $\sigma^x_{i,i\pm\hat{x}}$ and $\sigma^x_{i,i\pm\hat{y}}$ in terms of spin-orbital generators can be given. We emphasize, however, that the individual Ising electric field operators have no local representation in terms of gauge-invariant spin-orbital degrees of freedom.

Conversely, one may ask how spin-orbital operators such as $\tau^\alpha$, $s^\beta$ and $s^\alpha\tau^\beta$ are represented in terms $\sigma^x_{i,i\pm\hat{x}}$ and $\sigma^y_{i,i\pm\hat{y}}$. To this end, we first rewrite spin-orbital operators in terms of Majorana fermions according to Eq. (5) and then use the mapping in Eq. (19). Given the non-local nature of the string operator, we note that most spin-orbital operators lead to non-local expressions involving either the full string operator (as, e.g., in $s^x\tau^x = ib^1c^x$), or an extensive segment of the string operator (as, e.g., in $\tau^x = -ib^2b^3$). Given that these expressions only yield limited insight, we omit them here. However, special cases are $\tau^y = -ib^3_ib^1_i$ as well as $s^z\tau^y = ib^2_ib^4_i$. In those cases, the string operator cancels, and the result of applying Eq. (19) reads

$$
\tau^y_i = ib^1_ib^3_i = i\left(-\sigma^z_{i,i-\hat{y}}\right)\sigma^x_{i,i+\hat{x}}\sigma^x_{i,i-\hat{y}} = \sigma^x_{i,i+\hat{x}}\sigma^y_{i,i-\hat{y}},
\tag{24}
$$

for $i \in A$ sublattice. For $j \in B$, an analogous calculation leads to $\tau^y_j = \sigma^x_{j-\hat{x},j}\sigma^y_{j+\hat{y},j}$. Similarly, one obtains $s^z_i\tau^y_i = \sigma^x_{i,i-\hat{x}}\sigma^y_{i,i+\hat{y}}$ if $i \in A$ and $s^z_j\tau^y_j = \sigma^x_{j+\hat{x},j}\sigma^y_{j-\hat{y},j}$ for $j \in B$ sublattice. In Sec. 4 we will investigate in detail these terms as perturbations to the solvable lattice $\mathbb{Z}_2$ lattice gauge theory.

### 3.1.2 Correspondence between fermion parity constraint and $\mathbb{Z}_2$ Gauss law

The gauge-invariance of these link operators should be reflected in the Jordan-Wigner transformation of the local fermion parity constraint $\mathcal{D}_j$, which we aim to relate to the usual $\mathbb{Z}_2$ Gauss

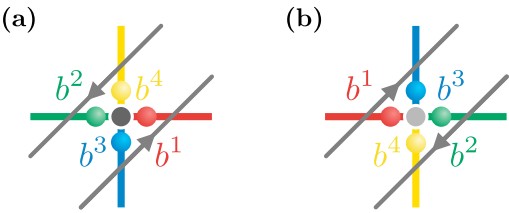

Figure 4: Illustration for ordering of gauge Majorana fermions along the Jordan-Wigner string for A [panel (a)] and B [panel (b)] sublattice sites.

law operator $G_j = (-1)^{f_j^\dagger f_j} \prod_{k \in +_j} \sigma_{jk}^x$ in the Wegner's formulation. Clearly, $-\mathrm{i}c^x c^y = (-1)^{f^\dagger f}$ in Eq. (6) straightforwardly carries over to the matter fermion parity in $G_j$, with $f = (c^x + \mathrm{i}c^y)/2$, so we can focus on the transformation of the remaing part of Eq. (6), the gauge Majorana parity $b_j^1 b_j^2 b_j^3 b_j^4$. We first consider $j \in$ A sublattice and reorder the fermion operators so that they are ordered along the string (see Fig. 4),

$$b_j^1 b_j^2 b_j^3 b_j^4 = -b_j^2 b_j^4 b_j^1 b_j^3. \tag{25}$$

We can now use the expressions of $b_i^\mu$ as given in Eq. (19), where the two string operators in the products $b_j^2 b_j^4$ and $b_j^1 b_j^3$ cancel except on bonds emanating site $j$,

$$b_j^2 b_j^4 = \left[ \prod_{l < \langle j, j-\hat{x}\rangle} (-\sigma_l^z) \right] \sigma_{j,j-\hat{x}}^x \left[ \prod_{l < \langle j, j+\hat{y}\rangle \equiv \langle j, j-\hat{x}\rangle - 1} (-\sigma_l^z) \right] \sigma_{j,j+\hat{y}}^x$$
$$= \sigma_{j,j-\hat{x}}^x \left[ \ldots \left( -\sigma_{j+\hat{y}+\hat{x},j+\hat{y}}^z \right) \left( -\sigma_{j,j+\hat{y}}^z \right) \right] \left[ \ldots \left( -\sigma_{j+\hat{y}+\hat{x},j+\hat{y}}^z \right) \right] \sigma_{j,j+\hat{y}}^x = -\mathrm{i}\sigma_{j,j-\hat{x}}^x \sigma_{j,j+\hat{y}}^y, \tag{26}$$

where we used that $\sigma^z \sigma^x = \mathrm{i}\sigma^y$ on the type-4 link $\langle j, j+\hat{y}\rangle$. Similarly, we have

$$b_j^1 b_j^3 = -\mathrm{i}\sigma_{j,j+\hat{x}}^x \sigma_{j,j-\hat{y}}^y, \tag{27}$$

so that the gauge Majorana parity on $j \in$ A sublattice sites is written as

$$b_j^1 b_j^2 b_j^3 b_j^4 = \sigma_{j,j-\hat{x}}^x \sigma_{j,j+\hat{y}}^y \sigma_{j,j+\hat{x}}^x \sigma_{j,j-\hat{y}}^y. \tag{28}$$

The same procedure can be repeated for $j \in$ B sublattice sites, where we find it convenient to re-order $b_j^1 b_j^2 b_j^3 b_j^4 = -b_j^3 b_j^1 b_j^4 b_j^2$. Performing the same analysis as above (using the prescription in Eq. (19)) for B sublattice sites, one obtains

$$b_j^1 b_j^2 b_j^3 b_j^4 = \sigma_{j+\hat{y},j}^y \sigma_{j-\hat{x},j}^x \sigma_{j-\hat{y},j}^y \sigma_{j+\hat{x},j}^x. \tag{29}$$

We therefore find that under the constructed mapping, the local fermion parity maps into a (slightly modified) $\mathbb{Z}_2$ Gauss law operator,

$$D_j = (-1)^{f_j^\dagger f_j} b_j^1 b_j^2 b_j^3 b_j^4 \mapsto \tilde{G}_j = (-1)^{f_j^\dagger f_j} \sigma_{j,j+\hat{x}}^x \sigma_{j,j-\hat{x}}^x \sigma_{j,j+\hat{y}}^y \sigma_{j,j-\hat{y}}^y. \tag{30}$$

Hence the transformed gauge constraint commutes with $\sigma^x$ on type-1(2) links, and $\sigma^y$ on type-3(4) links, i.e. $[\tilde{G}_j, \sigma_{j,j\pm\hat{x}}^x] = 0$ and $[\tilde{G}_j, \sigma_{j,j\pm\hat{y}}^y] = 0$, which precisely matches the results of the discussion in Sec. 3.1.1.

Let's transform the Gauss law now into the conventional form. Since the Hamiltonian in Eq. (14) is invariant under (local) U(1) rotations of the Pauli spinor $\sigma_{ij}^{\alpha}$ about the $\hat{z}$-axis, we are free to rotate

$$(\sigma^x, \sigma^y, \sigma^z) \mapsto (-\sigma^y, \sigma^x, \sigma^z), \text{ on all type-3 and type-4 links, i.e. } \langle i, i \pm \hat{y} \rangle. \tag{31}$$

We stress that inverting this step (i.e. rewriting $\sigma_{i,i\pm\hat{y}}^x$ as $\sigma_{i,i\pm\hat{y}}^y$) is crucial for the application of the inverse mapping, i.e. from Wegner's $\mathbb{Z}_2$ LGT formulation, which is conventionally written in terms of the gauge field $\sigma_{ij}^z$ and electric field operators $\sigma_{ij}^x$ on all links, to a Kitaev-type Majorana-based construction.

Under this unitary transformation, the Hamiltonian $\tilde{\mathcal{H}}_{\text{SOL}}$ [see Eq. (14)] remains invariant and is equivalent to $\mathcal{H}_{\text{LGT}}$ with $J = 2K$, $\mu = 0$ and $J_\square = 0$. The transformed constraint operator becomes

$$\tilde{G}_j \mapsto (-1)^{f_j^\dagger f_j} \prod_{l \in +_j} \sigma_l^x \equiv G_j, \tag{32}$$

which is equivalent to the conventional $\mathbb{Z}_2$ Gauss law constraint operator as given in Eq. (3). We have therefore found an explicit (invertible) mapping between both the Hamiltonians and the Gauss laws in conventional Wegner $\mathbb{Z}_2$ lattice gauge (coupled to dispersing fermionic matter) on the one side, and fermionic constructions of $\mathbb{Z}_2$ gauge theory on the other side, which naturally appear in the exact solution of Kitaev-type spin(-orbital) models.

## 3.2 Boundaries

While above discussion is concerned with a mapping between *bulk* degrees of freedom, we must further specify its action on boundary degree of freedom. Moreover, the discussion of the behaviour of the operators $\sigma_{ij}^x$ and $\sigma_{ij}^y$ defined in Eq. (18) under $\mathbb{Z}_2$ transformations by the local fermion parity $D_j$ was focused on transformations in the bulk and neglected transformations on the first bond of the Jordan-Wigner path.

We find it convenient to discuss these issues in a cylindrical geometry, matching the quasi-one-dimensional nature of the constructed transformation. To this end, we define a lattice of unit cells containing A and B sites, with lattice vectors $\boldsymbol{n}_1 = (1,1)^\top$ and $\boldsymbol{n}_2 = (-1,1)^\top$ with periodic boundary conditions in the $\boldsymbol{n}_2$ direction and open boundary conditions in the $\boldsymbol{n}_1$ direction with a "zigzag" boundary termination, as shown in Fig. 3.

### 3.2.1 Parity constraint and Gauss law at boundary

In the conventional $\mathbb{Z}_2$ gauge theory, it appears natural to define the $\mathbb{Z}_2$ Gauss law on these boundary sites as

$$G_i^{\text{boundary}} = (-1)^{f_i^\dagger f_i} \prod_{j \in \text{n.n.}(i)} \sigma_{i,j}^x, \tag{33}$$

where n.n.$(i)$ denotes the set of sites neighboring the boundary site $i$.

Considering the Kitaev-type fermionic construction, this implies that only two of the four gauge Majoranas at the boundary sites are used to form the $\mathbb{Z}_2$ gauge field $u_{ij} = i b_i^\mu b_j^\mu$, and the remaining two "dangling" Majorana degrees of freedom are degenerate. To remove this degeneracy, we can either only place a single spin (rather than a spin-orbital) degree of freedom at these boundary sites, thereby reducing the edge Hilbert space, or explicitly add gauge-invariant "mass" terms which gap out the dangling modes at each site by fixing their joint parity. For example, at the lower boundary in Fig. 3, we can add

$$\mathcal{H}_{\text{boundary}} \sim m \, i b_i^1 b_i^4 = m \left( 2 g_i^\dagger g_i - 1 \right), \tag{34}$$

where we define the complex fermion $g_i = \left(b_i^1 + ib_i^4\right)/2$. For $m > 0$, the ground state has that fermion empty, implying $ib_i^1 b_i^4 = -1$. Then, the local fermion parity constraint $D_i = +1$ at these lower boundary sites can be written

$$-ib_i^1 b_i^2 b_i^3 b_i^4 c_i^x c_i^y = +1 \quad \Longleftrightarrow \quad \overbrace{(-ib_i^1 b_i^4)}^{=+1} b_i^2 b_i^3 c_i^x c_i^y = +1$$

$$\Longleftrightarrow \quad (ib_i^2 b_i^3)(-1)^{f_i^\dagger f_i} = +1, \tag{35}$$

where the first equivalence makes explicit the correspondence to a local fermion parity constraint for the Majorana parton construction of a single $S = 1/2$ in terms of four Majorana fermions [40]. We now use the prescription of Eq. (18) for the gauge Majorana bilinear in Eq. (35), where $j \in$ B sublattice,

$$ib_j^2 b_j^3 = -ib_j^3 b_j^2 = -i\sigma_{j+\hat{y},j}^y \left[\ldots\left(-\sigma_{j+\hat{x}+\hat{y},j+\hat{y}}^z\right)\left(-\sigma_{j+\hat{x},j}^z\right)\right]\left[\ldots\left(-\sigma_{j+\hat{x}+\hat{y},j+\hat{y}}^z\right)\right]\sigma_{j+\hat{x},j}^y$$
$$= \sigma_{j+\hat{y},j}^y \sigma_{j+\hat{x},j}^x, \tag{36}$$

and analogous manipulations can be made at the top boundary, for $j \in$ A. With the rotation $(\sigma^x, \sigma^y) \mapsto (\sigma^y, -\sigma^x)$ on $\langle i, i \pm \hat{y}\rangle$ bonds as introduced earlier we therefore recover the conventional boundary Gauss law in Eq. (33).

### 3.2.2  Closed Jordan-Wigner string and global $\mathbb{Z}_2$ transformation

We have noted earlier that the string operator $\prod_{l<\langle ij\rangle}(-u_l)$ is invariant under gauge transformations acting on sites in the bulk of the string, i.e. for $l < \langle ij\rangle - 1$. It remains to be clarified how gauge transformations act on the first site ($i = 0$) marked by a purple star in Fig. 3. Given that the Jordan-Wigner string is closed in the cylinder geometry of Fig. 3, the concrete choice is a matter of convention. Nevertheless, the closed nature of the string leads to a subtlety that we will discuss now. To this end, we note that by expanding the product we can write

$$\sigma_{ij}^x = \left[(-u_{0,0-\hat{x}})(-u_{0-\hat{x}-\hat{y},0-\hat{x}})\ldots(-u_{(\langle ij\rangle-1)})\right]b_i^\mu, \tag{37}$$

and similarly for $\sigma_{ij}^y$ (in this section, we work with the un-rotated $\sigma^x, \sigma^y$ bond operators). As discussed earlier, the string $[\ldots]$ is invariant under $\mathbb{Z}_2$ gauge transformations induced by the fermion parity operator $D_k$ for any site $k$ except for the first site and last site of the string. The $\mathbb{Z}_2$ transformation at the last site is compensated by $b_i^\mu$ (on type-1(2) links for $\sigma^x$ and type-3(4) links for $\sigma^y$). However, acting with $D_0$, we use $D_0 u_{0,0-\hat{x}} = -u_{0,0-\hat{x}}D_0$ to argue that $D_0$ generates the transformation

$$\sigma_{i,i\pm\hat{x}}^x \mapsto s_0\sigma_{i,i\pm\hat{x}}^x, \quad \sigma_{i,i\pm\hat{y}}^y \mapsto s_0\sigma_{i,i\pm\hat{y}}^y, \ \forall\, i, \quad \text{and} \quad \sigma_{0,0-\hat{x}}^z \mapsto s_0\sigma_{0,0-\hat{x}}^z, \quad \sigma_{0,0-\hat{y}}^z \mapsto s_0\sigma_{0,0-\hat{y}}^z, \tag{38}$$

with $s_0 = \pm1 \in \mathbb{Z}_2$. This transformation therefore corresponds to a *global* $\mathbb{Z}_2$ transformation $(\sigma^x, \sigma^y, \sigma^z) \rightarrow (-\sigma^x, -\sigma^y, \sigma^z)$ (equivalent to conjugation $\sigma^a \mapsto \sigma^z \sigma^a \sigma^z$) paired with a *local* $\mathbb{Z}_2$ gauge transformation on the links emanating the site $i = 0$.

We can find the generator of this combined global and local symmetry operation in the conventional (bosonic) $\mathbb{Z}_2$ lattice gauge theory by using Eq. (18) for the local fermion parity operator $D_0$ at site $i = 0$ (with the $b^1, b^4$ modes gapped out by the protocol discussed in the previous section). We stress that periodic boundary conditions along $\mathbf{n}_2$ imply that $b_0^3$ involves the full Jordan-Wigner string (except for the last bond), see also Fig. 3. Explicitly, we have

$$D_0 = (-1)^{f_0^\dagger f_0}\left(ib_0^2 b_0^3\right) = -(-1)^{f_0^\dagger f_0}\left(ib_0^3 b_0^2\right) \tag{39}$$

$$= -(-1)^{f_0^\dagger f_0} \times i\sigma_{0,0-\hat{y}}^x \left[\left(-\sigma_{0,0-\hat{x}}^z\right)\ldots\left(-\sigma_{0-\hat{x}-\hat{y},0-\hat{y}}^z\right)\right]\sigma_{0,0-\hat{x}}^x$$

$$= (-1)^{f_0^\dagger f_0} \times \sigma_{0,0-\hat{y}}^y \left[\left(-\sigma_{0,0-\hat{x}}^z\right)\ldots\left(-\sigma_{0-\hat{x}-\hat{y},0-\hat{y}}^z\right)\left(-\sigma_{0,0-\hat{y}}^z\right)\right]\sigma_{0,0-\hat{x}}^x,$$

where in the last equality we have used $-i\sigma^x = \sigma^y(-\sigma^z)$ to complete the product over all bonds in the brackets. Under the mapping, the local fermion constraint operator thus becomes

$$D_0 \mapsto \tilde{G}'_0 = \underbrace{(-1)^{f_0^\dagger f_0}\sigma^x_{0,0-\hat{x}}\sigma^y_{0,0-\hat{y}}}_{\tilde{G}_0}\underbrace{\prod_l \sigma^z_l}_{G^z}, \tag{40}$$

where the product extends over all links $l$. We therefore find that the transformed fermion parity operator $D_0$ maps into a modified boundary Gauss law operator $\tilde{G}'_0$, which is a product of the usual $\mathbb{Z}_2$ boundary Gauss law $\tilde{G}_0$ and the generator $G^z$ of the *global* $\mathbb{Z}_2$ transformation $\sigma^{x,y} \to -\sigma^{x,y}$. Further note that $G^z$ commutes with both the bulk $\tilde{G}_i$ and $\tilde{G}_i^{\text{boundary}}$. Moreover, notice that despite starting with the *local* constraint $D_0$, we end up with a *global* factor $G^z$ in the conservation of $G^z \times \tilde{G}_0$ because the Jordan-Wigner mapping is *non-local*. We emphasize that neither $G^z$ nor $\tilde{G}_0$ must be separately conserved, but it is their product that defines the actual constraint in the $\mathbb{Z}_2$ gauge representation Eq. (11) of the spin-orbital liquid. We elaborate on this subtlety in Appendix B, where the global parity constraint is analyzed.

The global $\mathbb{Z}_2$ transformation generated by $G^z$ has important consequences for the conventional Wegner representation of the spin-orbital liquid in this geometry. In particular, the usual electric term $\sim \sigma^x$ cannot arrise in the Hamiltonian because it does not commute with the Gauss constraint $\tilde{G}'_0$ in Eq. (40). Curiously, the electric term can be made invariant under the constraint by multiplying the bulk electric operators $\sigma^x$ by operators that anticommute with the local transformation $\tilde{G}_0$.

## 4 Application: Anisotropic confinement

One can use our non-local mapping both ways: We can rewrite non-integrable[3] perturbations of the Kitaev-type spin(-orbital) models within a more familiar framework of the Wegner $\mathbb{Z}_2$ lattice gauge theory, where perturbations that induce vison dynamics have been studied before. Conversely, the mapping allows us to rewrite non-integrable perturbations of the Wegner $\mathbb{Z}_2$ gauge theories to the fermionic formulation, and then map onto local interactions in gauge-invariant spin(-orbital) models. Those can then be studied using established analytical and numerical machinery.

As a non-trivial application of our mapping, here we will investigate a "staircase" bilinear electric interactions of the Ising gauge field. We will show that in the spin-orbital formulation these perturbations correspond to local polarizing fields for the (spin-)orbital degrees of freedom. Due to the mapping introduced earlier, these two approaches are complementary and shed different light on confinement transition with a strongly anisotropic character.

### 4.1 Staircase electric interaction

As already discussed in Sec. 3.2.2, the ordinary linear electric interaction $\mathcal{H}_\Gamma = -\Gamma\sum_l \sigma^x_l$ is *not* allowed in the lattice gauge theory emerging from the spin(-orbital) models as it does not commute with the constraint operator $\tilde{G}'_0 = \tilde{G}_0 \times G^z$ as given in Eq. (40). Moreover, inspecting the mapping Eq. (18), we find that $\mathcal{H}_\Gamma$ becomes non-local in the fermionic formulation.

Therefore, let's consider instead the following gauge-invariant and non-integrable perturbation (in the bulk) given by two-body interactions of the gauge-field,

$$\tilde{\mathcal{H}}_g = -\sum_{i\in A}\left(g_1\sigma^x_{i,i+\hat{x}}\sigma^y_{i,i-\hat{y}} + g_2\sigma^x_{i,i-\hat{x}}\sigma^y_{i,i+\hat{y}}\right) - \sum_{i\in B}\left(g_1\sigma^x_{i-\hat{x},i}\sigma^y_{i+\hat{y},i} + g_2\sigma^x_{i+\hat{x},i}\sigma^y_{i-\hat{y},i}\right), \tag{41}$$

---

[3]i.e. those that do not conserve the plaquette operators $W_p$ and thus induce dynamics of the gauge fields.

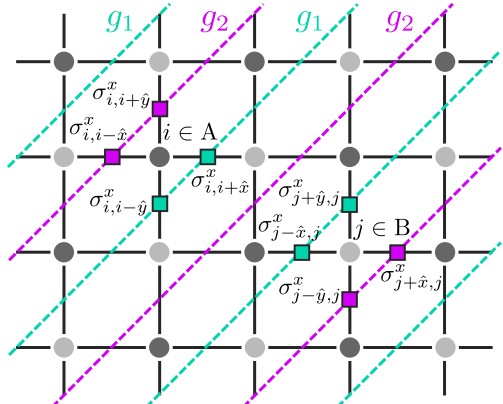

Figure 5: Illustration of the interaction $\mathcal{H}_g$ which couples gauge spins located on cornering links (represented by squares) to form diagonal ("staircase") XX chains (indicated by dashed lines), with alternating coupling constants $g_1$ (cyan) and $g_2$ (purple). Dark (light) grey circles indicate A (B) sublattice sites.

where the sums extends over all sites $i$ of the A and B sublattices. Here we again use the *bond-rotated reference frame* for $\sigma^\alpha$, such that the electric field operators correspond to $\sigma^x$ on type-1(2) links $\langle i, i \pm \hat{x} \rangle$, and $\sigma^y$ on type-3(4) links $\langle i, i \pm \hat{y} \rangle$, and the corresponding Gauss law given by Eq. (30). Note that in the conventional basis, where the electric field operators are given by $\sigma^x$ *on all bonds*, this interaction reads (using the redefinition Eq. (31) on $\tilde{\mathcal{H}}_g$),

$$\mathcal{H}_g = -\sum_{i \in \mathrm{A}} \left( g_1 \sigma^x_{i,i+\hat{x}} \sigma^x_{i,i-\hat{y}} + g_2 \sigma^x_{i,i-\hat{x}} \sigma^x_{i,i+\hat{y}} \right) - \sum_{i \in \mathrm{B}} \left( g_1 \sigma^x_{i-\hat{x},i} \sigma^x_{i+\hat{y},i} + g_2 \sigma^x_{i+\hat{x},i} \sigma^x_{i-\hat{y},i} \right), \quad (42)$$

which corresponds to an Ising-like interaction for the electric field along "staircases" as illustrated in Fig. 5. Note that these terms can be understood as spatially anisotropic gradients for the electric field [14]. In principle, each term in the sum in Eq. (42) could have different coupling strengths. Here to maintain translational invariance of the system (with respect to lattice vectors $\boldsymbol{n}_1$ and $\boldsymbol{n}_2$), we introduce two couplings $g_1$ and $g_2$ that control the interaction strengths on the two complementary staircases in Fig. 5.

We can now express the interaction $\tilde{\mathcal{H}}_g$ in the Kitaev-type Majorana representation. Note that for the application of our mapping, it is important that the indices $i, j$ in $\sigma^a_{i,j}$ are ordered such that $i \in \mathrm{A}$ and $j \in \mathrm{B}$. Using Eq. (18), the fermionic representation then reads

$$\tilde{\mathcal{H}}_g \mapsto -\left\{ \sum_{i \in \mathrm{A}} \left[ -g_1 b_i^1 \left( -u_{i,i-\hat{y}} \right) b_{i-\hat{y}}^3 - g_2 b_i^2 \left( -u_{i,i+\hat{y}} \right) b_{i+\hat{y}}^4 \right] \right.$$
$$\left. + \sum_{j \in \mathrm{B}} \left[ -g_1 b_{j-\hat{x}}^1 \left( -u_{j-\hat{x},j} \right) b_j^3 - g_2 b_{j+\hat{x}}^2 \left( -u_{j+\hat{x},j} \right) b_j^4 \right] \right\}, \quad (43)$$

where we have used that the overlap of the string operators in the products $\sim \sigma^x \sigma^y$ squares to one and thus cancels. One may now use $u_{i,i-\hat{y}} = i b_i^3 b_{i-\hat{y}}^3$ to write $u_{i,i-\hat{y}} b_{i-\hat{y}}^3 = i b_i^3 \left( b_{i-\hat{y}}^3 \right)^2 = i b_i^3$, and similarly for the other terms. We thus get a Hamiltonian with *site-local* bilinear interactions between different flavors of gauge Majoranas,

$$\tilde{\mathcal{H}}_g \mapsto -\sum_{i \in \mathrm{A,B}} \left( g_1 i b_i^1 b_i^3 + g_2 i b_i^2 b_i^4 \right), \quad (44)$$

where the sum extends over all sites $i$ in both sublattices. Using Eqs. (5) and (7), which imply $\tau^y = i b^1 b^3$ and $s^z \tau^y = i b^2 b^4$, the gauge-invariant Majorana bilinears can be rewritten in

terms of local orbital and spin-orbital operators, respectively,

$$\mathcal{H}_g^{\text{SOL}} = -\sum_i \left( g_1 \tau_i^y + g_2 s_i^z \tau_i^y \right). \tag{45}$$

Note that this result was also obtained by an inverse calculation (i.e. starting with the Majorana representation of the spin-orbital operators to obtain expressions in terms of $\sigma_{i,i\pm\hat{x}}^x$ and $\sigma_{i,i\pm\hat{y}}^y$ at the end of Sec. 3.1.1). The first term of this Hamiltonian has a natural interpretation in the spin-orbital language as a polarizing field for the orbital degree of freedom and may arise in physical systems with orbital degeneracies from pressure/strain [71]. The second term aims at aligning spin and orbital degrees of freedom at each site along a certain direction.

Finally, as shown earlier [44, 46], the Zeeman magnetization along $\hat{z}$ can be written (up to a constant) in terms of a (gauge-invariant) chemical potential for the matter fermion

$$\mathcal{H}_h = -h \sum_i s_i^z \mapsto -h \sum_i \left( -i c_i^x c_i^y \right) = h \sum_i \left( 2 f_i^\dagger f_i - 1 \right), \tag{46}$$

where the chemical potential $\mu = -2h$.

## 4.2 Warm-up: Anisotropic confinement in the absence of fermionic matter

To analyze the effect of the stair-case perturbation $\mathcal{H}_g$ on the deconfined phase, we first consider the pure $\mathbb{Z}_2$ gauge theory, where dynamical fermionic matter is absent. As will be shown here, by utilizing the Kramers–Wannier duality we can reduce the problem to a collection of transverse-field Ising chains along diagonals.

For simplicity, we will focus on the case of uniform couplings $g_1 = g_2 \equiv g$. To suppress fermions, we take the limit $\mu \to -\infty$, such that $f_i^\dagger f_i = 0 \ \forall i$. From the mapping in Eq. (46) it is clear that $\mu \to -\infty$ corresponds to a large Zeeman field $h \to \infty$ for the spin degrees of freedom. In this limit, the spin degrees of freedom become polarized, $\langle s^z \rangle = 1$, and only the orbitals $\tau_j$ remain as dynamical physical degrees of freedom.

### 4.2.1 Effective orbital Hamiltonian and matter-free gauge theory

For $h/K \gg 1$, we can derive an effective Hamiltonian for these orbital degrees of freedom. Projected to the spin-polarized sector $s^z = +1$ (corresponding to first-order perturbation theory), the Kitaev-type interaction $\mathcal{H}_{\text{SOL}}$ in Eq. (4) vanishes. The first non-trivial contribution arises at fourth-order perturbation theory [44], yielding a plaquette term

$$\mathcal{H}_\square^{\text{eff}} = -J_\square \sum_i \tau_i^x \tau_{i+\hat{x}}^y \tau_{i+\hat{x}+\hat{y}}^x \tau_{i+\hat{y}}^y, \tag{47}$$

which is identical to Wen's exactly solvable square lattice model [72], exhibiting (gapped) $\mathbb{Z}_2$ topological order. Note that Eq. (47) can be written $\mathcal{H}_\square^{\text{eff}} = -J_\square \sum_\square \mathbb{P}_{s^z=1} W_\square \mathbb{P}_{s^z=1}$ in terms of the plaquette operators Eqs. (12) and (13) projected to $s^z = +1$. The orbital-polarizing contribution $\sim g$ gives a non-trivial contribution to the effective Hamiltonian already at first-order perturbation theory,

$$\mathcal{H}_g^{\text{eff}} = \mathbb{P}_{s^z=1} \mathcal{H}_g \mathbb{P}_{s^z=1} = -2g \sum_i \tau_i^y. \tag{48}$$

Considering the Hamiltonian $\mathcal{H}_\square^{\text{eff}} + \mathcal{H}_g^{\text{eff}}$, it is clear that the topologically ordered phase present in the Wen plaquette model (for small $g \ll J_\square$) cannot be continously connected to the topologically trivial ground state at $g \gg J_\square$, where $\tau^y = 1$.

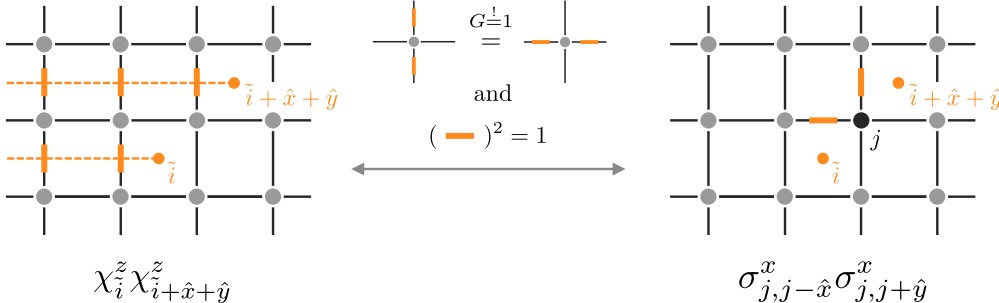

$$\chi^z_{\tilde{i}} \chi^z_{\tilde{i}+\hat{x}+\hat{y}} \qquad\qquad\qquad \sigma^x_{j,j-\hat{x}} \sigma^x_{j,j+\hat{y}}$$

Figure 6: Illustration of the duality mapping from electric field variables $\sigma^x_{ij}$ (indicated by bold orange lines) to the $\mathbb{Z}_2$ monopoles $\chi^z_{\tilde{i}}$ which live on the dual lattice of the square lattice. Here, we have chosen that the string in Eq. (50) extends from $\tilde{i}$ in the $-\hat{x}$ direction, but using the Gauss law $G_i = \prod_{l \in i} \sigma^x_l$, the path may be deformed arbitrarily.

We now discuss the same scenario in terms of the $\mathbb{Z}_2$ LGT (written in terms of Wegner's Ising gauge spins). In the absence of fermions, the corresponding Hamiltonian reads $\mathcal{H}^{\text{eff}}_{\text{LGT}} = \mathcal{H}_\Box + \mathcal{H}_g$ with

$$\mathcal{H}_\Box = -J_\Box \sum_\Box \prod_{\langle ij \rangle \in \Box} \sigma^z_{ij}, \tag{49}$$

and $\mathcal{H}_g$ given in Eq. (42). The gauge constraint (Gauss law) becomes $G_j = \prod_{l \in +_j} \sigma^x_l$. Clearly, for $J_\Box \gg g$, the system is in the deconfined phase of $\mathbb{Z}_2$ lattice gauge theory, whereas, as will be explained in the following, the ground state at $g \gg J_\Box$ approaches a trivial product state.

### 4.2.2 Confinement transition

To discuss the confinement transition from the topologically ordered phase to the confined product state, we first focus on the LGT formulation and perform a (bulk) Kramers–Wannier duality mapping of the $\mathbb{Z}_2$ lattice gauge theory (recall that there is no dynamical matter present). Working on the dual square lattice, (where plaquettes become sites), we define

$$\chi^x_{\tilde{i}} = \prod_{l \in \Box_{\tilde{i}}} \sigma^z_l, \quad \text{and} \quad \chi^z_{\tilde{i}} = \prod_{l \in \gamma(\tilde{i})} \sigma^x_l, \tag{50}$$

where $\tilde{i}$ denotes a site of the dual lattice, and $\Box_{\tilde{i}}$ corresponds to the square plaquette (on the original lattice) surrounding that dual lattice site, and $\gamma(\tilde{i})$ is some *arbitrary* path from infinity leading up to site $\tilde{i}$ of the dual lattice, bisecting links of the original lattice. Hence, $\chi^x_{\tilde{i}}$ is an operator which counts the $\mathbb{Z}_2$ flux through the plaquette $\tilde{i}$, while $\chi^z_{\tilde{i}}$ corresponds to a vison, a $\mathbb{Z}_2$ monopole operator with an attached t'Hooft line. Note that the precise path of $\gamma(\tilde{i})$ does not matter as long as the Gauss law $G_i = +1$ is strictly enforced.

In these dual variables, the plaquette term $\mathcal{H}^{\text{dual}}_\Box = -J_\Box \sum_{\tilde{i}} \chi^x_{\tilde{i}}$ amounts to a transverse field term, whereas the interaction $\mathcal{H}_g$ leads to diagonal (second-nearest-neighbor) interactions along the $\hat{x} + \hat{y}$-diagonal on the dual lattice,

$$\mathcal{H}^{\text{dual}}_g = -2g \sum_{\tilde{i}} \chi^z_{\tilde{i}} \chi^z_{\tilde{i}+\hat{x}+\hat{y}}, \tag{51}$$

see Fig. 6 for an illustration of the mapping. We therefore conclude that *in the bulk*, the dual

Hamiltonian to $\mathcal{H}_\square + \mathcal{H}_g$ decomposes into transverse-field Ising chains along the $\hat{x} + \hat{y}$ diagonal,

$$\mathcal{H}^{\mathrm{dual}} = -J_\square \sum_{\tilde{i}} \chi_{\tilde{i}}^x - 2g \sum_{\tilde{i}} \chi_{\tilde{i}}^z \chi_{\tilde{i}+\hat{x}+\hat{y}}^z \,. \tag{52}$$

These *decoupled* transverse field Ising chains can be solved *exactly* (for example, in terms of free fermions using a Jordan-Wigner transformation), and possess a second-order phase transition at $2g = J_\square$, with the transition lying in the two-dimensional classical Ising universality class. Within the dual description (in terms of $\chi^x, \chi^z$), the *polarized* phase with $\chi_{\tilde{i}}^x = +1$ corresponds to the deconfined phase of the $\mathbb{Z}_2$ lattice gauge theory, with excitations given by "spin flips" with $\chi_{\tilde{i}}^x = -1$ (corresponding to visons in the $\mathbb{Z}_2$ gauge theory), which cost energy $\Delta E = 2J_\square$ for $J_\square \gg g$. On the other hand, at $g/J_\square \gg 1$, the dual model has a two-fold degenerate "ferromagnetic" ground state with $\chi_{\tilde{i}}^z = \pm 1 \ \forall \tilde{i}$, corresponding to the confining phase, with excitations given by domain walls along the staircase chain in the $\chi_{\tilde{i}}^z$ basis, costing energy $\Delta E = 4g$ for $g \gg J_\square$. Indeed, inspecting $\mathcal{H}_g$ in the lattice gauge theory formulation, it is clear that the ground state for large $g/J_\square$ consists of aligned $\sigma^x$ along one-dimensional staircases.

Since the Hamiltonian $\mathcal{H}_{\mathrm{LGT}}^{\mathrm{eff}}$ is invariant under $\sigma^a \mapsto \sigma^y \sigma^a \sigma^y$ (for $a = x, y, z$), which in particular takes $\sigma^x \mapsto -\sigma^x$, there appears to be a two-fold degeneracy per staircase pattern. However, we stress that the ground state at large $g/J_\square$ *does not* possess a (sub-)extensive degeneracy in the cylinder geometry of Fig. 3. This is because the staircases are coupled together by the boundary Gauss laws, for example

$$G_j = \sigma_{i,i+\hat{x}}^x \sigma_{i,i+\hat{y}}^x \equiv +1 \,, \tag{53}$$

at the boundary. An explicit analysis of the boundary in the dual model in Eq. (52) is cumbersome due to the non-locality of the Kramers–Wannier duality mapping. We instead consider the original formulation in terms of the Ising electric operators $\sigma^x$: Since $\sigma_{i,i+\hat{x}}^x$ and $\sigma_{i,i+\hat{y}}^x$ belong to distinct diagonal staircases, this implies the $\sigma^x = +1$ or $\sigma^x = -1$ along the two neighboring lines cannot be picked independently, but rather all chains become locked together.[4] Thus only a global two-fold degeneracy is left.

In a complementary viewpoint, one may directly consider the spin-orbital Hamiltonian $\mathcal{H}_\square^{\mathrm{eff}} + \mathcal{H}_g^{\mathrm{eff}}$ given in Eqs. (47) and (48). This model is precisely the Wen-plaquette model perturbed by a longitudinal field as studied in Ref. [73], where a duality mapping was used to show that the phase transition at $J_\square = 2g$ lies in the $(1+1)$-dim. Ising universality class. This matches precisely the results that we have obtained within the $\mathbb{Z}_2$ lattice gauge theory description.

### 4.2.3 Ground-state degeneracy at $g \gg J_\square$

One remaining question concerns the degeneracy of the ground state at large $g/J_\square \gg 1$. Inspecting $\mathcal{H}_g$ as given in Eq. (42) leads to the conclusion that for large $g$, the ground state is two-fold degenerate (i.e. a ferromagnet for the $\sigma_{ij}^x$ variables). However, previous analysis of the spin-orbital model found a unique ground state in this parameter regime, adiabatically connected to the trivial product state $\prod_j |\tau_j^y = +1\rangle$.

This apparent contradiction is resolved by explicitly following through with the mapping on a cylindrical geometry from the spin-orbital model to the standard $\mathbb{Z}_2$ LGT formulation, as

---

[4]We emphasize that even though the chains are not decoupled, but rather locked together by the boundary Gauss laws, the *bulk* analysis in terms of the dual variables is still applicable, and in particular the result that there is a second-order phase transition at $2g = J_\square$ in the $(1+1)$-dim. Ising universality class still holds.

described in Sec. 3. In particular, the Gauss law on the first site of the constructed Jordan-Wigner string leads to the constraint (compare Eq. (40)),

$$G^z \times \tilde{G}_0 = \left(\prod_l \sigma_l^z\right) \times \left((-1)^{f_0^\dagger f_0}\, \sigma_{0,0-\hat{x}}^x \sigma_{0,0+\hat{y}}^x\right) \overset{!}{=} +1\,, \tag{54}$$

which we did not account for in our earlier analysis of the perturbed lattice gauge theory. Since we work in the limit $-\mu \sim h \to \infty$, where fermions are absent, we can set $f_0^\dagger f_0 = 0$ and thus $(-1)^{f_0^\dagger f_0} = 1$ in Eq. (54). Enforcing the remaining parts of above constraint is equivalent to projecting to the subsector where we have $G^z \times \tilde{G}_0 = +1$. To this end, we introduce a projection operator $\mathbb{P}^z = (\mathbb{1} + G^z \tilde{G}_0)/2$. Applying $\mathbb{P}^z$ to the two degenerate "ferromagnetic" states $|+\rangle = \prod_l |\sigma_l^x = +1\rangle$ and $|-\rangle = \prod_l |\sigma_l^x = -1\rangle$ at large $g/J_\square \gg 1$ gives the same cat state (up to normalization and a global phase)

$$\mathbb{P}^z |+\rangle \sim \mathbb{P}^z |-\rangle \sim (|+\rangle + |-\rangle)\,, \tag{55}$$

where we use that $\tilde{G}_0 |\pm\rangle = +|\pm\rangle$. Hence, the ground state in the confining phase of the spin-orbital problem is unique after properly accounting for gauge invariance at the boundary, matching the analysis of the spin-orbital model.

## 4.3 Anisotropic confinement with dynamical fermionic matter

Here we will investigate how the phenomenon of anisotropic confinement triggered by the staircase electric interaction in Eq. (42) is affected by dynamical itinerant fermion matter charged under the Ising gauge theory. In this section we switch off $J_\square$, but instead focus on the coupling between gauge fields and fermions.

### 4.3.1 Perturbing the orbital sector: $g_1 \neq 0$ and $g_2 = 0$

We find it instructive to discuss first the case of $g_1 \neq 0$ and $g_2 = 0$ in Eqs. (42) and (45). Compared to the generic case, the model enjoys a particle-hole symmetry which acts as $\mathsf{s}^z \to -\mathsf{s}^z$ in the spin sector and does not affect the orbital degrees of freedom. We will concentrate our attention on the symmetric point with $S^z = \sum_i \mathsf{s}_i^z = 0$.

In this case, in the limit $g_1 \to \infty$ the term $\mathcal{H}_g = -\sum_i g_1 \tau_i^y$ imposes a ground state where the orbital degrees of freedom are fully polarized, amounting to a confining phase of the $\mathbb{Z}_2$ lattice gauge theory. For $g_1 \gg J$, we can obtain an effective Hamiltonian in first order perturbation theory by projecting the spin-orbital Hamiltonian $\mathcal{H}_{\text{SOL}}$ into the subspace with $\tau_i^y = +1\ \forall i$, yielding (recall that we use $\mathsf{s}^\alpha$ to denote Pauli matrices rather than $S = 1/2$ operators)

$$\mathbb{P}_{\tau^y=+1}\mathcal{H}_{\text{SOL}}\mathbb{P}_{\tau^y=+1} = -K\sum_{\langle ij\rangle_2}\left(\mathsf{s}_i^x \mathsf{s}_j^x + \mathsf{s}_i^y \mathsf{s}_j^y\right) - K\sum_{\langle ij\rangle_4}\left(\mathsf{s}_i^x \mathsf{s}_j^x + \mathsf{s}_i^y \mathsf{s}_j^y\right)\,. \tag{56}$$

This effective Hamiltonian thus corresponds to spin-1/2 degrees of freedom with XY exchange interaction along "staircases chains" in the $\hat{x} + \hat{y}$ direction.

Owing to its essentially one-dimensional nature, this effective Hamiltonian can be exactly solved in terms of spinless fermions $c, c^\dagger$ by means of a Jordan-Wigner transformation with $\mathsf{s}_j^z = 2c_j^\dagger c_j - 1$ and $\sigma_j^+ = e^{-i\sum_{i<j} c_j^\dagger c_j} c_j^\dagger$, $\sigma_j^- = (\sigma_j^+)^\dagger$, where $i$ and $j$ index sites along the staircase chain. We can then write

$$\mathbb{P}_{\tau^y=+1}\mathcal{H}_{\text{SOL}}\mathbb{P}_{\tau^y=+1} = -2K\sum_{i\in A}\left(c_i^\dagger c_{i+\hat{x}} + c_{i+\hat{x}}^\dagger c_{i+\hat{x}+\hat{y}} + \text{h.c.}\right)\,. \tag{57}$$

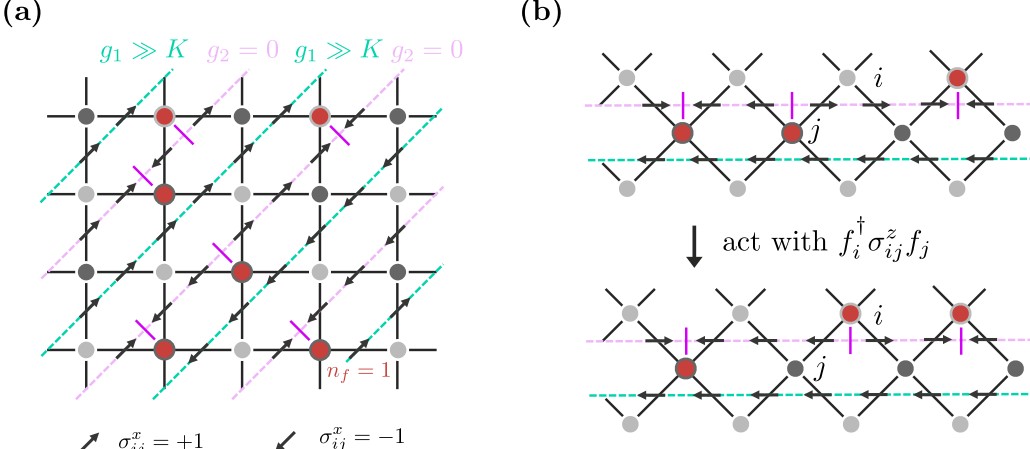

Figure 7: (a) For $g_1 \gg K$ and $g_2 \equiv 0$, every second staircase chain (in cyan) possesses Ising-type ferromagnetic order, while the gauge spins on the remaining diagonals (light purple) are free to fluctuate. Excitations correspond to domain walls on the chain (indicated by perpendicular purple lines). Because of the $\mathbb{Z}_2$ Gauss law, an odd number of $\sigma^x = -1$ around a given site must be accompanied by a fermion (depicted in red), thus binding fermions to domain walls on the purple chains. (b) In first order perturbation theory in $\mathcal{H}_{\text{LGT}} \sim f_i^\dagger \sigma_{ij}^z f_j$, a dispersion for fermion-domain wall objects is generated. Note that the generated dispersion at this order is one-dimensional along the light purple chains.

Diagonalizing this effective Hamiltonian yields the spectrum $E(q) = -4K\cos(q)$ with momenta $q \in [-\pi, \pi]$ as usual. We conclude that in the orbital-polarized state, the first-order effective model is composed of decoupled half-filled fermionic chains along the north-east diagonal.

How can we understand all that in the $\mathbb{Z}_2$ lattice gauge theory formulation? Here, the perturbation can be rewritten as

$$\mathcal{H}_{g_1} = -g_1 \sum_{i \in A} \left( \sigma_{i,i+\hat{x}}^x \sigma_{i,i-\hat{y}}^x + \sigma_{i,i+\hat{x}}^x \sigma_{i+\hat{x},i+\hat{x}+\hat{y}}^x \right), \tag{58}$$

which can be seen in the bulk to correspond to decoupled ferromagnetic chains of $\sigma_{ij}^x$ gauge spins along every second north-east staircase of the lattice. Note that $g_2 = 0$ implies that there are no interactions on the other set of north-east diagonals of the lattice. As expected, the perturbation in Eq. (58) is particle-hole symmetric since it commutes with the particle-hole transformation that we introduced in Sec. 2.1. We now consider the limit $g_1 \gg K$, where the ground state is expected to correspond to ferromagnetic order on the staircase chains, induced by $\mathcal{H}_{g_1}$. Then, excitations correspond to domain walls on top of this ferromagnetic order which cost energy $\Delta E = 2g_1$. Note that there is a second set of staircase chains, as shown in Fig. 7, where the Ising gauge spins are free to fluctuate. On these staircases, domain walls should not cost any energy. However, we stress that in addition to energetic considerations, domain walls as excitations are only allowed if they satisfy the Gauss law in Eq. (3). Since the Gauss law of each site involves two gauge spins located on the same $g_1$-chain (and thus their product equals $+1$), this implies that domain walls on the $g_2$-chains must bind a fermion for gauge invariance (see also illustration in Fig. 7). Working at a fixed partial filling for the fermions, we find that for each of the "free" chains there is a large degeneracy associated with placing the domain wall-fermion pairs at different sites. Clearly, this degeneracy is lifted in first order perturbation theory in $J/g_1 \ll 1$: We note that $\mathcal{H}_{\text{LGT}} = -J \sum_{\langle ij \rangle} f_i^\dagger \sigma_{ij}^z f_j$ has a non-trivial action on bond $\langle ij \rangle$ if either site $i$ or $j$ are occupied by fermion. In this case, it acts by flipping the gauge

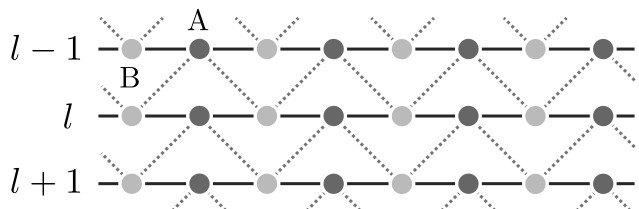

Figure 8: Illustration of interchain coupling: Staircase chains (formed by type-2 and type-4 links) obtained in first-order perturbation theory in $K/g_1 \ll 1$ are displayed "flattend" horizontally. Dashed lines correspond to interchain couplings obtained from second-order perturbation theory.

spin $\sigma_{ij}^x \mapsto -\sigma_{ij}^x$ and hopping the fermion from site $i$ to $j$, or vice versa. By the previous argument, a fermion can only be located at site $i$ or $j$ if there is a domain wall located at that site. Flipping the gauge spin then corresponds to moving the domain wall by one site along the chain, and the explicit fermion hopping operators ensure that the fermion moves together with the domain wall, see Fig. 7 b, such that again the local Gauss laws are satisfied. As a result, at first order perturbation theory in $J/g_1 \ll 1$ an effective one-dimensional dispersion along the chains is generated, consistent with the insights gained from the gauge-invariant spin-orbital formulation.

Now we are ready to go beyond first-order perturbation theory. In the spin-orbital language, the fermionic chains become coupled by the perturbing Hamiltonian, given by the terms $\mathcal{H}^{\text{SOL}}$ on the $\mu = 1, 3$-type links. In order to return to the ground state with $\tau_i^y = +1$, the only non-trivial processes in second order perturbation theory are given by the application on $\mathcal{H}^{\text{SOL}}$ on the same bond, such that the effective Hamiltonian becomes

$$\mathcal{H}_{\text{eff}}^{(2)} = -\frac{K^2}{2g_1} \sum_{\langle ij \rangle_{1,3}} \left( \mathbb{1} - s_i^z s_j^z \right), \tag{59}$$

where the intermediate state (with $\tau_i^y = \tau_j^y = -1$ on the two endpoints of the bond) costs energy $4g_1$, and we have used that $\left( s_i^x s_i^x + s_i^y s_j^y \right)^2 = 2\mathbb{1} - 2s_i^z s_j^z$.

We thus find that the XY chains become coupled via repulsive density-density interactions with "two-to-one" interactions between individual sites as shown in Fig. 8. To study the system of weakly coupled chains, we note that after Jordan-Wigner transformation, the fermionic problem can be written in the form (here, $i$ labels unit cells with sites A,B and fermion occupation is measured with respect to half-filling)

$$\mathcal{H} = \frac{J_{\parallel}}{2} \sum_{i \in \text{u.c.}} \sum_{l \in \text{chains}} \left( c_{i,A,l}^\dagger c_{i,B,l} + c_{i,B,l}^\dagger c_{i+1,A,l} + \text{h.c.} \right)$$
$$+ J_\perp \sum_{i \in \text{u.c.}} \frac{1}{2} \sum_{l \in \text{chains}} \left( n_{i,A,l} n_{i,B,l+1} + n_{i,B,l} n_{i,A,l-1} + n_{i,A,l} n_{i-1,B,l+1} + n_{i,B,l} n_{i+1,A,l-1} \right), \tag{60}$$

where $J_{\parallel} = -4K$ and $J_\perp = K^2/(2g_1)$, and the factor of $1/2$ in the second summand is to avoid double-counting the interchain interactions. We now employ "chain mean-field theory" [74, 75], where we use a mean-field approximation of the interaction term to decouple the individual chains and obtain a single-chain Hamiltonian. Note that in the case at hand, the resulting single-chain problem can be solved exactly. To be specific, we make the mean-field approximation

$$n_{i,A,l} n_{i,B,l+1} \rightarrow \langle n_{i,A,l} \rangle n_{i,B,l+1} + n_{i,A,l} \langle n_{i,B,l+1} \rangle - \langle n_{i,A,l} \rangle \langle n_{i,B,l+1} \rangle. \tag{61}$$

We find that a staggered mean-field parameter $\langle n_{i,A,l} \rangle = -\langle n_{i,B,l} \rangle = m \neq 0$ opens up a finite gap in the chain, as the corresponding order wavevector $\pi$ coincides with the Fermi wavevector of the half-filled chains, analogous to the Peierls instability. By solving the self-consistency equations for $m$ in the weak-coupling limit (see Appendix C for details), we find an exponentially small gap of the form

$$m \approx \frac{\Lambda}{2(J_\perp/|J_\parallel|)} e^{-\frac{\pi}{2(J_\perp/|J_\parallel|)}}, \tag{62}$$

where $\Lambda$ is some UV cutoff, and $J_\perp/J_\parallel = K/(8g_1)$. We therefore expect that the coupled chains become gapped, and the system develops a staggered fermion density order, implying antiferromagnetic Ising order in the out-of-plane spin component, $s_i^z \sim (-1)^i$. We emphasize, however, that the gap is exponentially small in $K/g_1$, which is required to be small for our perturbative analysis to be valid. Higher order perturbative inter-chain couplings might change qualitatively our result. A detailed study of the true nature of the confinement phase is left for a further study.

### 4.3.2 Perturbing both spin and orbital sectors

While in the previous subsection, we have focused on perturbing the orbital sector with a term $\sim g_1 \tau_i^y$ and demonstrated anisotropic confinement, we now discuss more general cases where also $g_2 \neq 0$ in Eq. (45) and finite Zeeman fields $h \neq 0$ in Eq. (46).

**General case.** We first comment on switching on some finite $g_2 \neq 0$ in Eq. (45), mostly focusing on the confined phase, where $g_1 \gg K$. We first note that if $g_2 \ll g_1$, we can still project $H_g^{\text{SOL}}$ onto the $\tau^y = +1$ sector, such that the perturbation $-g_2 s^z \tau^y$ amounts to an effective field for the XY chains, corresponding to an effective chemical potential for the Jordan-Wigner fermions (recall that $s_i^z = 2c_i^\dagger c_i - 1$) in Eq. (57),

$$\mathbb{P}_{\tau^y=+1} \mathcal{H}_{\text{SOL}} \mathbb{P}_{\tau^y=+1} = -2g_2 \sum_i c_i^\dagger c_i. \tag{63}$$

We further comment that for $g_1 \sim g_2 \gg K$, the ground state must be adiabatically connected to the product state $\prod_i |s_i^z = +1\rangle |\tau_i^y = +1\rangle$. This is a confining and gapped phase of matter. The nature of the phase transition from the spin-orbital liquid to this state, and possible intervening phases, is an interesting subject left for further study: In the presence of $g_1 \neq 0$ and $g_2 \neq 0$ the Kitaev model no longer possesses particle-hole symmetry, and thus the spin-orbital liquid can develop a finite magnetization $\langle s^z \rangle \neq 0$. Upon increasing $g_1 \sim g_2$, the system will then undergo a confinement transition (either of first or second order). The nature of the confined phase will depend on the ratio of $g_1$ and $g_2$: For $g_1 \gg g_2$ and non-zero $K$, one might expect the emergence of gapless chains (away from half-filling because Eq. (63) acts as an effective chemical potential), while for sufficiently large $g_2$ will give rise to a fully gapped (trivial) product state. Interchain couplings at higher-order perturbation theory may however substantially change these expectations, requiring a more careful analysis.

**Degenerate point.** From the preceeding discussion and Eq. (42), we note that in the case of $g_1 = g_2 \equiv g$, adding an arbitrary long electric string along a staircase chain corresponds to placing two domain walls along a $\sigma^x \sigma^x$ chain (see Fig. 5). This comes with an energy cost of $2 \times 2g$. Gauge invariance demands that an electric field string has fermionic charges at its endpoints. If there is a finite chemical potential $\mu > 0$, the associated energy cost of the string with charges is $\Delta E = 4g - 2\mu$. We hence note that in the special case with $g_1 = g_2$ and $g \equiv \mu/2$, there is *no energy cost* associated with adding electric strings along the staircase chains.

This special point with a large degeneracy can also be characterized in the spin-orbital language. Consider the purely local terms in the spin-orbital Hamiltonian in Eqs. (45) and (46),

$$\mathcal{H}_g^{\text{SOL}} + \mathcal{H}_h = -\sum_i \left( g_1 \tau^y + g_2 s_i^z \tau_i^y + h s_i^z \right). \tag{64}$$

Diagonalization of the Hamiltonian reveals that for $h < 0$ and $g_1 = g_2 = \pm|h|$, the local ground state is threefold degenerate. By recalling that the chemical potential relates to the Zeeman field $h$ as $\mu = -2h$, this matches precisely the regime where electric strings do not cost any energy.

We can obtain an effective model that lifts this extensive degeneracy ($\sim 3^{N_{\text{sites}}}$) by projecting the Hamiltonian of the spin-orbital liquid $\mathcal{H}_{\text{SOL}}$ into the subspace of locally three-fold degenerate states, yielding (see Appendix D for details)

$$\mathcal{P}\mathcal{H}_{\text{SOL}}\mathcal{P} = -K \sum_{\langle ij \rangle_{1,3}} \left( L_i^x L_j^x + Q_i^{yz} Q_j^{yz} \right) - K \sum_{\langle ij \rangle_{2,4}} \left( L_i^y L_j^y + Q_i^{xz} Q_j^{xz} \right), \tag{65}$$

where $L^\alpha$ denote spin-1 operators, and $Q^{\alpha\beta} = \{L^\alpha, L^\beta\}$ denote corresponding quadrupolar operators.

The interactions among the dipolar components resemble the spin-1 $90°$ compass model [76–78], although here the bond-dependence of interactions refers to the respective staircase chains in the lattice. The Hamiltonian further contains bond-dependent interactions in the quadrupolar channel, which – to our knowledge – have not been discussed previously, but bear similarities to the spin-1 quadrupolar Kitaev model on the honeycomb lattice, recently discussed by Verresen and Vishwanath in Ref. [79]. The ground state and phase diagram of this effective model, which evades an exact solution, is an interesting subject that we leave for further study.

## 5    Conclusion

In the present work, we related two well-known incarnations of the Ising $\mathbb{Z}_2$ lattice gauge theory coupled to fermionic matter. Specifically, we have constructed an explicit Jordan-Wigner mapping between Wegner's formulation, which is written in terms of Ising gauge spins placed on bonds, and Kitaev-type construction, where the gauge field is built from Majorana partons with the gauge constraint given by the local fermion parity. We envision that the mapping could be generalized to $\mathbb{Z}_N$ gauge theories.

As an application, we considered exactly-solvable $\nu = 2$ spin-orbital liquids on the square lattice and analyzed perturbations that spoil integrability of the model and eventually induce confinement. Based on complementary analyses in the spin-orbital language as well as $\mathbb{Z}_2$ lattice gauge theory formulation, we have shown that the confinement phase is strongly anisotropic. In the absence of fermionic matter (corresponding to a spin-polarized orbital liquid), the problem is mapped onto a collection of transverse-field Ising chains and therefore the confinement transition is in the one-dimensional quantum Ising universality class. On the other hand, in the presence of fermionic matter, the nature of the confined phase and the corresponding confinement transition remains to be further clarified. Maybe embedding the Ising gauge theory into the parent continuous compact U(1) lattice gauge theory coupled to a double-charge Higgs field and fermions can shed new light on those questions. This will likely involve significant additional technical efforts and is deferred to the future.

Beyond these immediate applications to spin-orbital liquids, the mapping may also be generalized. It would be interesting attempt extending the mapping to $\mathbb{Z}_N$ lattice gauge theories

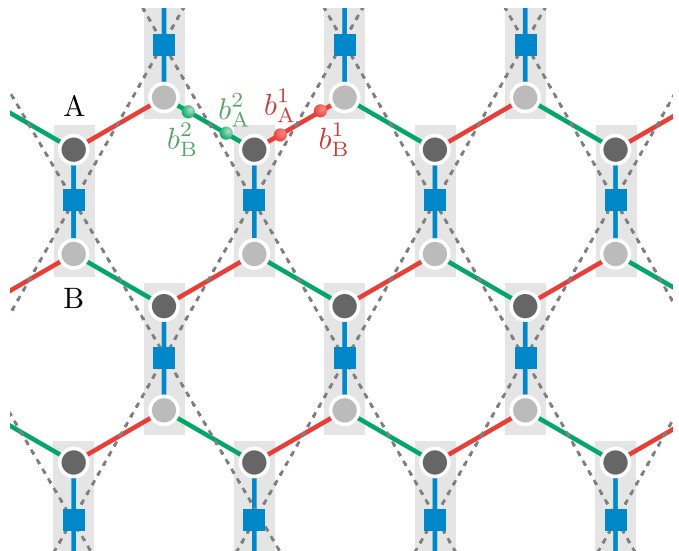

Figure 9: Illustration of mapping Kitaev's honeycomb model to a $p$-wave paired square lattice model. Contracting the type-3 (blue) links to points gives an effective (dashed) square lattice with sites indicated by blue squares. The Majorana fermions $b_{i,A}^1, b_{i,B}^1, b_{i,A}^2, b_{i,B}^2$ are seen to be equivalent to the "gauge" Majoranas in the solution of the spin-orbital model in Eq. (4).

coupled to matter (and potentially their $N \to \infty$ U(1) limit). This will open new perspectives on quantum phases and dynamics of those lattice gauge theories as effective models for quantum matter.

## Acknowledgments

We gratefully acknowledge discussions with L. Balents, U. Borla, T. Grover, and H.-H. Tu.

**Funding information** This work was supported by the Deutsche Forschungsgemeinschaft (DFG, German Research Foundation) through a Walter Benjamin fellowship, Project ID 449890867 (UFPS). S.M. is supported by Vetenskapsrådet (grant number 2021-03685), Nordita and STINT. This work was performed in part at Aspen Center for Physics, which is supported by National Science Foundation grant PHY-2210452.

## A  Kitaev honeycomb model as a $\mathbb{Z}_2$-gauged p-wave superconductor on a square lattice

Our mapping does not work on the honeycomb geometry for Kitaev's original $S = 1/2$ model, governed by the Hamiltonian

$$\mathcal{H} = -K \left[ \sum_{\langle ij \rangle_1} \mathsf{s}_i^x \mathsf{s}_j^x + \sum_{\langle ij \rangle_2} \mathsf{s}_i^y \mathsf{s}_j^y + \sum_{\langle ij \rangle_3} \mathsf{s}_i^z \mathsf{s}_j^z \right]. \tag{A.1}$$

The reason is that the Jordan-Wigner covering bonds pass each vertex three times, implying that the string cannot cancel.

However, following the work by Chen and Nussinov[5] [80], we can map the Kitaev honeycomb model to a complex fermion dispersing on the square lattice coupled to a $\mathbb{Z}_2$ gauge field. To this end, we first employ the standard Kitaev parton construction $s^\alpha = ib^\alpha c$ with constraint $D = b^x b^y b^z c = 1$. Explicitly keeping track of the sublattice indices, we can write

$$\mathcal{H} = K \sum_{i \in \text{u.c.}} \sum_{\delta = n_1, n_2, 0} iu_i(\delta) c_{i,A} c_{i+\delta,B}, \tag{A.2}$$

where $n_{1,2} = (\pm 1, \sqrt{3})^\top / 2$ are the lattice vectors of the honeycomb lattice, and $u_i(\delta) = ib^\alpha_{i,A} b^\alpha_{i+\delta,B}$ is the $\mathbb{Z}_2$ gauge field written in terms of the gauge Majoranas.

We now introduce a complex fermion $f_i = (c_{i,A} + ic_{i,B})/2$ on each unit cell, and introduce a square lattice by contracting the unit cells into sites as depicted in Fig 9. We can form a combined constraint for each unit cell by consider the product of the two fermion parities,

$$D_{i,A} D_{i,B} = \left( ib^1_{i,A} b^1_{i,B} \right) \left( ib^2_{i,A} b^2_{i,B} \right) \left( ib^3_{i,A} b^3_{i,B} \right) \left( ic_{i,A} c_{i,B} \right) \tag{A.3}$$

$$= b^1_{i,A} b^1_{i,B} b^2_{i,A} b^2_{i,B} u_i(0)(-1)^{f^\dagger_i f_i} \overset{!}{=} 1, \tag{A.4}$$

where we have used $ic_{i,A} c_{i,B} = -(-1)^{f^\dagger_i f_i}$ as usual. Without loss of generality, we can fix the gauge $u_i(0) = +1$. Then, the constraint $D_i \equiv D_{i,A} D_{i,B} = 1$ is seen to be equivalent to the total fermion parity constraint on the square lattice, Eq. (10), as obtained in the solution of the spin-orbital model, where $b^1_{i,A}, \ldots, b^2_{i,B}$ can be appropriately relabelled to $b^1_i, \ldots b^4_i$.

The conserved plaquette operators on the honeycomb lattice,

$$W_p = \prod_{l \in \partial p} u_l, \tag{A.5}$$

are easily seen to correspond to the plaquette operators $W_\square$ on the square lattice upon using the gauge fixing $u_i(0) = +1$.

Finally, the Hamiltonian becomes

$$\mathcal{H} = K \sum_i \sum_{a = \hat{x}, \hat{y}} u_{i,i+a} \left( f^\dagger_i f_{i+a} + f^\dagger_i f^\dagger_{i,a} + \text{h.c.} \right) + K \sum_i \left( 2f^\dagger_i f_i - 1 \right), \tag{A.6}$$

where the sum $i$ now extends over sites of the square lattice, and $\hat{x}, \hat{y}$ refer to the unit lattice vectors of the square lattice. We hence find that the Kitaev honyecomb model is transformed to complex fermions with $p$-wave pairing on the square lattice, coupled to the $\mathbb{Z}_2$ gauge field $u_{ij}$. It is now straighforward to apply our Jordan-Wigner mapping from Sec. 3 to this formulation.

A crucial difference to Eq. (11) as obtained from the spin-orbital lattice model is that the fermions do not enjoy a global U(1) symmetry due to the presence of pairing terms. Indeed, as elucidated in Refs. [44, 46], the global U(1) symmetry of the fermions $f_j \mapsto e^{i\varphi} f_j$ in Eq. (11) is equivalent to the built-in spin U(1) spin rotation symmetry about the $\hat{z}$-axis in Eq. (4). On the other hand, Kitaev's $S = 1/2$ model, Eq. (A.1), does *not* possess such a symmetry.

# B  Global parity constraints in different formulations of Ising gauge theory coupled to complex fermion matter

Here we compare global parity constraints in the Wegner's and Kitaev's formulations of the Ising gauge theory coupled to fermion matter in the finite cylinder geometry of Fig. 3.

---

[5]However, in their work, the fermion is coupled to a *classical* $\mathbb{Z}_2$ field, rather than a *gauge* field.

We start from the Wegner's formulation, where the product of all $\mathbb{Z}_2$ Gauss laws as given in Eq. (3) reads

$$1 = \prod_j G_j = \prod_j \left[ (-1)^{f_j^\dagger f_j} \prod_{l \in +_j} \sigma_l^x \right] \equiv \prod_j (-1)^{f_j^\dagger f_j} = D_f^{\text{tot}}, \qquad \text{(B.1)}$$

where we use in the last equality that the product over all sites emanating all sites $j$ covers each link twice, and $(\sigma^x)^2 = 1$. This implies that in the conventional bosonic formulation of $\mathbb{Z}_2$ gauge theory with Gauss-law constraints [Eq. (3)], the total matter fermion parity must be fixed to unity (all physical states contain even number of $\mathbb{Z}_2$ charged fermions), but there are *a priori* no further constraints on the global gauge field parity $G^z = \prod_l \sigma_l^z$.

We now contrast this with the corresponding Kitaev's fermionic formulation that is relevant for spin-orbital liquids. Consider the product over all local constraints, corresponding to the total fermion parity

$$\prod_j D_j = \prod_i b_i^1 b_i^2 b_i^3 b_i^4 \times \prod_j \left( -\mathrm{i} c_j^x c_j^y \right) = \pm \left[ \prod_{l = \langle mn \rangle} u_l \right] \times \prod_j (-1)^{f_j^\dagger f_j} \equiv D_u^{\text{tot}} \times D_f^{\text{tot}}, \quad \text{(B.2)}$$

where we use that we can rearrange the product of *all* Majoranas of the system into a product $D_u^{\text{tot}}$ of the $\mathbb{Z}_2$ gauge field $u_l$ on all links $l$ and the total matter fermion parity $D_f^{\text{tot}}$. Enforcing the constraint that the local fermion parity $D_j = +1 \forall j$ implies that the RHS of Eq. (B.2) is equal to unity. These considerations are straightforwardly carried over to the conventional (bosonic) formulation of $\mathbb{Z}_2$ lattice gauge theory upon identifying $u_l$ with $\sigma^z$, thus finding that $G^z \times D_f^{\text{tot}} = 1$. So neither the matter fermion parity $D_f^{\text{tot}}$, nor the global gauge field parity $G^z$ are separately conserved, only their product is.

## C  Details on chain mean-field theory

Here, we consider the Hamiltonian in Eq. (60) with the mean-field approximation of the inter-chain interactions as in Eq. (61), where by translational symmetry we take $\langle n_{i,A,l} \rangle = m_A$ and $\langle n_{i,B,l} \rangle = m_B$. The single-chain Hamiltonian then attains the form

$$\mathcal{H}_l = \frac{J_\parallel}{2} \sum_{i \in \text{u.c.}} \left( c_{i,A}^\dagger c_{i,B} + c_{i,B}^\dagger c_{i+1,A} + \text{h.c.} \right) + \frac{J_\perp}{2} \sum_i \left( 4 n_{i,A} m_B + 4 n_{i,B} m_A - 4 m_A m_B \right). \qquad \text{(C.1)}$$

Note that $\mathcal{H}_l$ is a free-fermion Hamiltonian, so that we can obtain its spectrum exactly: In momentum space, the Hamiltonian is then written as[6]

$$H = \sum_k \underline{c}_k^\dagger \begin{pmatrix} 2J_\perp m_B & \frac{J_\parallel}{2} \left( 1 + \mathrm{e}^{-\mathrm{i}k} \right) \\ \frac{J_\parallel}{2} \left( 1 + \mathrm{e}^{\mathrm{i}k} \right) & 2J_\perp m_A \end{pmatrix} \underline{c}_k - 2J_\perp N_{\text{u.c.}} m_A m_B. \qquad \text{(C.2)}$$

We now notice that a gap opens if $m_B \neq m_A$. We let $m_B = -m_A \equiv m$ and then the spectrum is given by

$$\varepsilon_\pm(k) = \pm \sqrt{4 J_\perp^2 m^2 + \frac{J_\parallel^2}{2} \left( 1 + \cos k \right)}. \qquad \text{(C.3)}$$

---

[6]For convenience we have chosen lattice constants such that there is a spacing of $2a = 1$ between two unit cells, corresponding to $a = 1/2$ with $a$ denoting the spacing between two sites on the chain.

Observe that for $J_\perp m \equiv 0$, the two bands correspond to the dispersion of the $\sim \cos k$ band backfolded onto half of the original Brillouin zone. The mean-field state energy per chain and per site is given by

$$E_0/N_{\text{u.c.}} = \langle \mathcal{H} \rangle / N_{\text{u.c.}} = -\int_{-\pi}^{\pi} \frac{dk}{2\pi} \sqrt{4J_\perp^2 m^2 + \frac{J_\parallel^2}{2}(1 + \cos k) + 2J_\perp m^2} \,. \tag{C.4}$$

A mean-field saddlepoint is found if $\partial E_0 / \partial m = 0$, which yields the trivial solution $m \equiv 0$, or the self-consistency equation (we shift $k \to k + \pi$ in the integrand but choose the domain of integration $k \in [-\pi, \pi]$, which is allowed by periodicity of the integrand)

$$\frac{1}{J_\perp} = \int_{-\pi}^{\pi} \frac{dk}{2\pi} \frac{1}{|J_\parallel| \sqrt{4(J_\perp/J_\parallel)^2 m^2 + \frac{1}{2}(1 - \cos k)}} \,. \tag{C.5}$$

We can solve this implicit equation for $m$ only numerically. However, note that for $J_\perp m \to 0$, the right-hand side diverges because of the singularity at $k = 0$. To extract the asymptotic behaviour, we expand in small $k$ (adding some UV cutoff $\Lambda$), yielding

$$\frac{1}{J_\perp/|J_\parallel|} = \int_0^{\Lambda} \frac{dk}{\pi} \frac{1}{\sqrt{4(J_\perp/J_\parallel)^2 m^2 + \frac{k^2}{4}}} \,. \tag{C.6}$$

For small $m$, we now expand the denominator, $\sqrt{4(J_\perp/J_\parallel)^2 m^2 + \frac{k^2}{4}} \approx \frac{k}{2} + \frac{4m^2(J_\perp/J_\parallel)^2}{k} + \ldots$ such that the resulting integral (with some redefined cutoff) becomes

$$\int_0^{\Lambda} \frac{k\,dk}{\frac{k^2}{2} + 4m^2(J_\perp/J_\parallel)^2} = \int_{4m^2(J_\perp/J_\parallel)^2}^{(\Lambda')^2} \frac{du}{u} = 2\log\left(\frac{\Lambda'}{2m(J_\perp/|J_\parallel|)}\right) \,. \tag{C.7}$$

Solving Eq. (C.6) for $m$, we then obtain

$$m \approx \frac{\Lambda'}{2(J_\perp/|J_\parallel|)} e^{-\frac{\pi}{2(J_\perp/|J_\parallel|)}} \,, \tag{C.8}$$

concluding that any $J_\perp > 0$ is sufficient to induce an instability of the XY chains and open up a (exponentially small) gap.

## D Derivation of effective Hamiltonian at threefold-degenerate point

Focusing on the case where $g_1 = g_2 = -|h|$, the onsite spin-orbital Hamiltonian in Eq. (64) can be written as

$$\mathcal{H}_{\text{loc}} = |h| \sum_i \mathcal{O}_i, \quad \text{with} \quad \mathcal{O}_i = \tau_i^y + s_i^z \tau_i^y + s_i^z, \tag{D.1}$$

where we note that the operator $\mathcal{O}_i$ satisfies $\mathcal{O}_i^2 = 3\mathbb{1} + 2\mathcal{O}_i$. This implies that we can define a projection operator $\mathcal{P}_i$ to the local threefold-degenerate subspace as

$$\mathcal{P}_i = \frac{3}{4}\mathbb{1} - \frac{1}{4}\mathcal{O}_i. \tag{D.2}$$

We can now use this operator for first-order perturbation theory, where we project the Kitaev-type spin-orbital exchange interactions in $\mathcal{H}_{\text{SOL}}$, Eq. (4), to the local threefold-degenerate

Table 1: Local spin-orbital operators projected into the three-fold degenerate ground state on each site, rewritten in terms of spin-1 generators operators $L, Q$ after a unitary basis change $U$.

| Spin-orbital operator $\mathcal{O}^{\mathrm{SO}}$ | $s^x\tau^x$ | $s^x\tau^y$ | $s^x\tau^z$ | $s^x\tau^0$ | $s^y\tau^x$ | $s^y\tau^y$ | $s^y\tau^z$ | $s^y\tau^0$ |
|---|---|---|---|---|---|---|---|---|
| $U^\dagger \mathcal{P}\mathcal{O}^{\mathrm{SO}}\mathcal{P} U$ | $L^x$ | $Q^{xz}$ | $-Q^{yz}$ | $-Q^{xz}$ | $-Q^{yz}$ | $-L^y$ | $-L^x$ | $L^y$ |

subspaces. The resulting Hamiltonian will involve interactions between the projected spin-orbital operators $\mathcal{P}_i s_i^\alpha \tau_i^\beta \mathcal{P}_i$ at different sites. These can be rewritten in terms spin-1 operators $L^\alpha$ (with $\alpha = x, y, z$) that act on the three-dimensional subspaces, with the explicit matrix representation $[L^\alpha]_{\beta\gamma} = -i\epsilon^{\alpha\beta\gamma}$. One can also introduce the five local quadrupolar operators $Q^{xy} = \{L^x, L^y\}, Q^{yz} = \{L^y, L^z\}, Q^{xz} = \{L^x, L^z\}$ as well as $Q^{x^2-y^2} = (L^x)^2 - (L^y)^2$ and $Q^{3z^2-r^2} = (3(L^z)^2 - 2\mathbb{1})/\sqrt{3}$, which together with the $L^\alpha$ form a complete basis of the eight Hermitian operators acting on a three-dimensional Hilbert space. We also find it convenient to perform a basis change after the projection to an eigenbasis of $\mathcal{O}_i$ and $\mathcal{H}_{\mathrm{loc}}$, with the explicit representation

$$U = \frac{1}{\sqrt{2}} \begin{pmatrix} 0 & 0 & 1 \\ 0 & 0 & -i \\ 1 & 1 & 0 \\ -i & i & 0 \end{pmatrix}. \tag{D.3}$$

We can then decompose the basis-rotated projected spin-orbital operators $U^\dagger \mathcal{P} s^\alpha \tau^\beta \mathcal{P} U$ in terms of the spin-1 operators $L^\alpha, Q^{\cdots}$, exploiting their orthonormality with respect to the trace $\mathrm{tr}[X_i, X_j]/2 = \delta_{i,j}$. We list the resulting expressions for the projected spin-orbital operators in Table 1.

With these effective spin-1 operators, it is now straightforward to write down the projected Kitaev spin-orbital Hamiltonian $\mathcal{H}_{\mathrm{SOL}}$, given by Eq. (65).

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
