# Peer review of "Wegner's Ising gauge spins versus Kitaev's Majorana partons: Mapping and application to anisotropic confinement in spin-orbital liquids"

_SciPost Physics, doi:SciPost Phys. 16, 147 (2024)_

## Round 1 · Referee Report · Anonymous (Referee 1) · 2023-10-30

Report

The authors bridges the gap between the Wegner’s Z2 lattice gauge theory with bosonic matter, and the Kitaev-like spin-orbital model with fermionic matter. Even though the spin-orbital model and the fermionization scheme has been uncovered in some of the author’s previous work, a direct comparison with the Wegner’s lattice gauge model remains elusive in the literature. In particular, the electric field is a fundamental degree of freedom in the lattice gauge theory, which fluctuates the gauge flux and drives a versatile of gauge phenomena such as confinement. However, it is less discussed in the Kitaev-like spin-orbital model, where people either consider a solvable model with static gauge flux, or deal with the Kitaev model beyond solvable limit only in the spin basis. In this report, the authors show a two-step mapping: (spin-orbital basis) —> (fractionalized Majorana basis) —> (bosonic lattice gauge basis). As an application, the authors discuss the anisotropic confinement consequence from a natural spin-orbital interaction, treating that from the pure gauge limit to the matter-gauge coupling scenario.

Based on my estimate, their work could appeal to readers from both the frustrated magnetism and lattice gauge theory community. Despite the limited discussion of gauge phenomena, they have opened an avenue for more discussions of the gauge theories in spin-orbital liquids. Sufficient details are provided in the writing. Overall, I recommend this manuscript to be published by SciPost Physics.

In addition, I leave the following questions for the authors:

  1. Eq.18 expresses the electric field sigma^{x(y)} in terms of the fractionalized Majorana fermion b. But since the interests originate from the spin-orbital model, how is the electric field expressed in terms of the original spin-orbital degrees of freedom (s, tau), which is by definition automatically a gauge invariant basis?

  2. Relatedly, for the natural perturbation or solvability breaking fields of (s, tau) in different angles, how are they expressed in the gauge language?

  3. While the authors have discussed the confinement transition, equally interesting is the Higgs transition from the gauge theory perspective: the Higgs transition from deconfined Z2 to trivial phase; and the un-Higgs transition from Z2 to deconfined U(1) state. The latter used to be observed in the lattice gauge theory, the toric code or the Kitaev honeycomb model beyond solvable limit. I wonder whether the authors can comment or relate their formalism to these transitions, charting out a guiding landscape for the gauge phases of matter in the spin-orbital liquids.

  • validity: -
  • significance: -
  • originality: -
  • clarity: -
  • formatting: -
  • grammar: -

Author:  Urban Seifert  on 2024-05-03  [id 4468]

(in reply to Report 1 on 2023-10-30)
Category:
remark
answer to question

We are grateful for the Referee's reading of our manuscript and are happy that they recommend our manuscript to be published in SciPost Physics. We have made several edits in response to their questions:

  1. and 2. We have provided a detailed explanation for expressing the electric-field operators in the $\mathbb{Z}_2$ gauge theory in terms of spin-orbital in Sec. 3.1: we find that $\sigma^x$ and $\sigma^y$ do not admit simple closed-form expressions in terms of the spin-orbital degrees of freedom, however pairs of such electric field operators along appropriate bonds map onto spin-orbital degrees of freedom (see, e.g., Eq. (24)).

  2. The Referee raises highly interesting open questions: Indeed, the Higgs transition from U(1) to $\mathbb{Z}_2$ lattice gauge theory could be studied by developing a U(1) lattice gauge theory with appropriate deformations. Its continuum limit could lead to new insights into the appropriate critical phenomena. By studying the deformed U(1) LGT that mimics the effect of finite couplings $g_1$ and $g_2$ (see Eq. (41)) might also allow novel insights into the critical theory of the deconfinement transition driven by these couplings. We further comment that our work can also be extended the $\mathbb{Z}_N$ lattice gauge theories, where we expect that one might construct a mapping from a "Wegner-type" theory (with clock and shift matrices placed on bonds) to a (para-)fermionic theory by means of generalized Jordan-Wigner transformations, and a subsequent rewriting in terms of a generalized spin-orbital theory. As these require significant technical efforts, we leave these topic for future study.

---

## Round 1 · Referee Report · Anonymous (Referee 2) · 2023-11-10

Strengths

  1. Clearly written.

  2. Of interest to community of lattice gauge theories and quantum magnetism.

Weaknesses

  1. Despite the methods not particularly novel, there is very limited acknowledgement and citations to precedents in the literature.

Report

The authors discuss a map between Wegner Z2 gauge theory and the Majorana parton representation of a spin orbital liquid model. The map is a Jordan-Wigner transformation between the gauge degrees of freedom, which are Ising spins in the Wegner Z2 gauge theory and the gauge Majorana bilinears. The authors apply the map to study an anisotropic confinement transition. The results are interesting and relevant for the to community of lattice gauge theories and quantum magnetism. Therefore I recommend publication.

However the paper has a very limited acknowledgement and missing citations to a series of previous works in the literature dealing with closely related constructions where dual representations of gauge theories are constructed with Jordan Wigner transformations. Below are some suggestions to improve the references.

  • Their Ref. 51 has related ideas (but it is only cited as dealing with something "more involved").

The following references (not cited) discuss or use related constructions/methods:

-S. B. Bravyi and A. Y. Kitaev, Annals of Physics 298, 210 (2002). - F. Verstraete and J. I. Cirac, Journal of Statistical Mechanics: Theory and Experiment 2005, P09012 (2005). - M. Levin and X.-G. Wen, Physical Review B 73, 035122 (2006). - R. C. Ball, Phys. Rev. Lett. 95, 176407 (2005). - Y.-A. Chen and A. Kapustin, Physical Review B 100, 245127 (2019). - D. Radicevic, arXiv preprint arXiv:1809.07757 (2018). - Y.-A. Chen Phys. Rev. Research 2, 033527 (2020). - H. C- Po, arXiv:2107.10842 (2021). - P. Rao and I. Sodemann, Phys. Rev. Research 3, 023120 (2021). - K. Li and H. Chun Po, Phys. Rev. B 106, 115109 (2022). -W. Cao, M.o Yamazaki, and Y. Zheng, Phys. Rev. B 106, 075150 (2022). - C. Chen, P. Rao, and I. Sodemann, Phys. Rev. Research 4, 043003 (2022). - L Goller, I. S. Villadiego, arXiv:2309.13116 (2023). - Y.-A. Chen and Y. Xu, PRX Quantum 4, 010326 (2023).

Requested changes

Improving the acknowledgement and citations to precedents in the literature.

  • validity: good
  • significance: good
  • originality: ok
  • clarity: good
  • formatting: good
  • grammar: good

Author:  Urban Seifert  on 2024-05-03  [id 4467]

(in reply to Report 2 on 2023-11-10)
Category:
remark
correction

We thank the Referee for their feedback and overall positive assessment of our work. We regret having neglected referencing previous constructions that may appear to be of similar nature. In the resubmitted manuscript, we have added several key references. However, we stress that our work is distinct from many previous contributions in the fact that we do not construct two-dimensional bosonization (i.e., a mapping between an ungauged fermionic problem and a $\mathbb{Z}_2$ gauged spin-1/2 model), but rather our key achievement consists in relating to common constructions of $\mathbb{Z}_2$ gauge theories (of "Kitaev"- and "Wegner"-type) to each other.

---

## Round 3 · Referee Report · Anonymous (Referee 1) · 2024-5-3

Report

Acceptance criteria met. Recommended for publication.

Recommendation

Publish (meets expectations and criteria for this Journal)

---

## Round 3 · Referee Report · Anonymous (Referee 2) · 2024-5-7

Report

The authors have taken into account the comments from my previous review. I recommend therefore publication.

Recommendation

Publish (meets expectations and criteria for this Journal)

---

## Round 3 · Author Response

We would like to thank the Editor for considering our paper for publication in SciPost.

We are grateful for the Referees' careful reading of our manuscript and their constructive comments. We are happy that their overall assessment of our work has been positive. We address their individual comments/concerns by replying to their reports, and provide a list of changes with our resubmission.

We hope that our resubmitted manuscript can be accepted for publication in SciPost in the present form.

Best regards,

Urban Seifert and Sergej Moroz

---

## Round 3 · List of Changes

Major changes to the text and content are marked in blue.

  • Added a number of references, in particular including Refs. 20, 22, 33, 34, 47-65.

  • In Sec. 3.1.1: Discussion on deriving expressions of gauge-invariant lattice gauge theory operators \sigma^x_{i,j} and \sigma^y_{i,j} in terms of spin-orbital operators, and vice versa.

  • In Sec. 4.1: Added sentences that references Sec. 3.1.1.

  • In Sec. 5: Added outlook/suggestions for studying deconfinement and Higgs transitions within a continuum critical field theory

---

## Editorial Decision

published